# Overexpression of *DfRaf* from Fragrant Woodfern (*Dryopteris fragrans*) Enhances High-Temperature Tolerance in Tobacco (*Nicotiana tabacum*)

**DOI:** 10.3390/genes13071212

**Published:** 2022-07-07

**Authors:** Chunhua Song, Qi Fan, Yuqing Tang, Yanan Sun, Li Wang, Mingchu Wei, Ying Chang

**Affiliations:** College of Life Sciences, Northeast Agricultural University, Harbin 150030, China; chunhuasong@163.com (C.S.); cyqfqfw@163.com (Q.F.); yuqingtang20010103@163.com (Y.T.); 13684603690@163.com (Y.S.); lxgirl12345@163.com (L.W.); cc09190087@163.com (M.W.)

**Keywords:** fragrant woodfern, carbon assimilation, high-temperature stress, *DfRaf*

## Abstract

Heat stress seriously affects medicinal herbs’ growth and yield. Rubisco accumulation factor (Raf) is a key mediator regulating the activity of ribulose-1,5-bisphosphate carboxylase/oxygenase (Rubisco), which plays important roles in carbon assimilation and the Calvin cycle in plants. Raf has been studied in many plants, but has rarely been studied in the important medicinal plant fragrant woodfern (*Dryopteris fragrans*). The aim of this study was to analyze the effects of Raf on carbohydrate metabolism and the response to heat stress in medicinal plants. In this study, high temperature treatment upregulated the expression of *DfRaf*, which was significantly higher than that of phosphoribokinase (*DfPRK*), Rubisco small subunits (*DfRbcS*), Rubisco large subunits (*DfRbcL*) and Rubisco activase (*DfRCA*). The subcellular localization showed that the DfRaf proteins were primarily located in the nucleus; DfPRK, DfRbcS, DfRbcL and DfRCA proteins were primarily located in the chloroplast. We found that overexpression of *DfRaf* led to increased activity of Rubisco, RCA and PRK under high-temperature stress. The H_2_O_2_, O_2_^−^ and MDA content of the DfRaf-OV-L2 and DfRaf-OV-L6 transgenic lines were significantly lower than those of WT and VC plants under high-temperature stress. The photosynthetic pigments, proline, soluble sugar content and ROS-scavenging ability of the DfRaf-OV-L2 and DfRaf-OV-L6 transgenic lines were higher than those of WT and VC plants under high-temperature stress. The results showed that overexpression of the *DfRaf* gene increased the Rubisco activity, which enhanced the high-temperature tolerance of plants.

## 1. Introduction

Plants can grow and develop normally within a certain temperature range. Temperatures higher (heat stress) or lower (cold stress) than the optimum temperature required for plant growth can inhibit or delay the growth [1,2,3,4]. Temperature is one of the most important natural determinants of plants’ photosynthetic activity, and over 90% of plant dry weight is derived from photosynthesis [5]. High temperatures can adversely impact photosynthetic rates, energy metabolism, the chloroplast’s submicroscopic structural integrity, photosynthetic pigment content and other key biochemical processes necessary to plant survival [6,7]. High temperatures not only cause changes in photosynthetic pigments and transpiration, but also substantially alter the activities of Calvin cycle-related enzymes [8,9], including ribulose-1,5-bisphosphate carboxylase/oxygenase (Rubisco), which is the key carboxylase involved in C3 plants’ photosynthetic processes [10]. Rubisco is a hexadecameric complex composed of eight large subunits (RbcL, 50–55 kDa) and eight small subunits (RbcS, 12–18 kDa), denoted as RbcL8S8 [11]. RbcL subunits are arranged as a tetramer of antiparallel RbcL dimers so that RbcL8 serves as the primary catalytic core, with eight RbcS subunits that dock at the top and bottom to stabilize RbcL8 and tune the activity of Rubisco [12]. Rubisco is a key enzyme in the process of plants’ carbon assimilation. Its function is to catalyze the carboxylation of RuBP. However, Rubisco’s catalytic efficiency is very low. It interacts with various phosphate sugars and lose its activity. For example, the substrate RuBP itself is a strong inhibitor of Rubisco [13]. Therefore, the activated state of Rubisco is often an important factor determining the rate of carbon assimilation. In plants, the activation of Rubisco depends on the action of Rubisco-activating enzyme (RCA). RCA releases the inhibitor bound to Rubisco by using the energy of ATP hydrolysis and changes the conformation of Rubisco, thereby stimulating its activity [14]. In wheat (*Triticum aestivum*), significant increases in RCA activity enable Rubisco to maintain high CO_2_ assimilation efficiency [15]. As a molecular chaperone of Rubisco [16], RCA can control and stabilize Rubisco activity and thereby regulate photosynthesis under high-temperature stress [17]. In *Zea mays*, another RbcL-interacting chaperone termed Rubisco accumulation factor (Raf) was identified [11,18]. Raf1 has been shown to promote recombinant Rubisco assembly in tobacco (*Nicotiana tabacum*) chloroplasts. Its overexpression can bolster Rubisco protein expression and activation [19]. Phosphoribokinase (PRK) is a critical regulator of carbon assimilation and controls the homeostatic balance between assimilation and regeneration activities [20]. PRK can catalyze ribulose-1,5-bisphosphate (RuBP) regeneration while providing a receptor for CO_2_ assimilation [21]. In studies where Rubisco levels in tobacco plants were decreased, reduced photosynthesis and growth were observed [22,23], with similar results also being found in rice plants [24]. Conversely, increases in the activity levels of Rubisco resulted in increased photosynthesis and biomass in tobacco plants, both under controlled conditions [25] as well as in the field under elevated CO_2_ [26]. In food crops such as tomato, increased RCA activities have been reported to result in similar increases in photosynthesis and growth, as well as improved chilling tolerance [27]. This indicates that the increase in carbon assimilation-related enzyme activity promotes the enhancement of plant photosynthesis and then promotes plant growth and yield, especially in important food crop varieties. At the same time, these results provide evidence that increasing the activity of carbon assimilation-related enzymes enhances high-temperature tolerance in plants. However, the relationship between carbohydrate metabolism and the response to high-temperature stress needs to be studied further.

Fragrant woodfern (*Dryopteris fragrans*) is an annual medicinal herb that grows in rock crevices and under wood [28], primarily in Wudalianchi City, Heilongjiang Province, China [29]. This region can experience significant temperature variability, with record high and low temperatures of 36.5 °C and −41.0 °C, respectively [30]. Fragrant woodfern plants contain benzene triphenols, terpenes, flavonoids and phenylpropanoid compounds that exhibit inhibitory effects on fungi. These plants have also been used to alleviate skin itching, allergies and rheumatoid arthritis [31]. However, habitat reductions and excess utilization have led wild fragrant woodfern to become increasingly endangered. Further studies of the molecular mechanisms governing fragrant woodfern’s stress resistance will therefore be beneficial for the artificial cultivation of this plant in order to meet consumer demand. Carbon assimilation is a key facet of the heat stress response pathways in plants. Therefore, fragrant woodfern is a valuable example for studying the heat stress tolerance function of carbon assimilation-related genes in ferns. We explored changes in carbon assimilation enzyme activity (including the Rubisco activation state, total Rubisco activity, initial Rubisco activity and RCA and PRK activity) and assessed the relative levels of the genes associated with carbon assimilation including *DfPRK, DfRbcS*, *DfRbcL*, *DfRCA* and *DfRaf* via quantitative real-time PCR (qRT-PCR). We isolated and identified a *DfRaf* gene from fragrant woodfern. Ectopic expression of *DfRaf* enhanced heat stress tolerance in tobacco (*Nicotiana tabacum*). These results demonstrate that *DfRaf* plays a positive role in heat resistance, which may provide a reference for breeding heat-resistant cultivars in future genetic engineering experiments.

## 2. Materials and Methods

### 2.1. Plant Materials and Treatment

Fragrant woodfern (*Dryopteris fragrans*) seedlings were procured from the laboratory of plant resources and molecular biology of Northeast Agricultural University. They were grown in containers containing a 1:1 mixture of vermiculite and grass charcoal soil at 25 ± 2 °C with a 16 h light–8 h dark photoperiod. One hundred and eighty pots of 12-week-old seeds (60 in each of the 3 replicates) with consistent growth were then transferred to a 42 °C growth room to simulate high-temperature stress conditions, respectively [32]. At 0, 0.5, 1, 3, 6, 12, 24 and 48 h after treatment at these temperatures, leaves were collected from these plants, snap-frozen and stored at −80 °C.

Tobacco seeds (*Nicotiana tabacum*) were surface-sterilized for 3 min in 75% ethanol, rinsed with sterile water and then germinated in 1/2 MS medium in a greenhouse with a 16 h light–8 h dark photoperiod at 25 ± 2 °C. Following germination, the seedlings were transferred to a flowerpot containing 1:1 (*V/V*) highly nutritious soil and vermiculite, and then cultured in a growth chamber with a 16 h light–8 h dark photoperiod at 25 ± 2°C.

### 2.2. Rubisco Activity Measurements

Initial and total Rubisco activity levels were measured as described previously [33]. Briefly, an enzyme solution was used for extraction by grinding 0.2 g of a fresh sample that had been snap-frozen with liquid nitrogen, after which 6 mL of the chilled extraction solution (50 mM Hepes-KOH (pH 7.5), 10 mM MgCl_2_, 10mM DTT, 2 mM EDTA, 10% glycerol (*w/v*), 1% BSA (*w/v*), 1% TritonX-100 (*v/v*) and 1.5% PVPP (*w/v*)) was added, and the mixture was ground to yield a homogenate. A 1 mL volume of this homogenate was then spun down for 10 min at 14,000 rpm at 4 °C, and the supernatants were immediately collected and used to analyze enzymatic activity.

Initial Rubisco activity was assessed by using a 900 μL volume of the reaction solution (100 mM *N*,*N*-bis (2-hydroxyethyl) glycine (pH 8.0), 20 mM magnesium chloride, 25 mM potassium bicarbonate, 3.5 mM ATP, 5mM creatine phosphate, 5 U 3-phosphorus acid glyceraldehyde dehydrogenase, 5 U 3-phosphoglycerate phosphokinase, 17.5 U phosphocreatine kinase and 0.25 mM NADH) to which 50 μL of the enzyme extract was added, followed immediately by the addition of 50 μL of RuBP (0.6 mM). The solution was then mixed via pipetting and the absorbance at 350 nm was quantified every 10 s for 2 min. 

Total activity was assessed by adding 50 μL of the enzyme to 900 μL of the reaction medium, after which Rubisco was activated via a 15 min incubation at 25 °C. Next, 50 μL of RuBP was added, and the absorbance at 340 nm was assessed every 10 s for 2 min.
(1)$$\mathrm{Rubisco}\;\mathrm{carboxylase}\;\mathrm{activity}\;(\mu \mathrm{mol}\cdot {\mathrm{mg}}^{-1}\cdot h^{-1})=\frac{\left(Eo-E1\right)\times Vt}{\left(2d\times \zeta \times \Delta t\times Vs\times m\right)}$$
where *Eo* and *E1* correspond to the A340 values before and after the reaction, *Vt* is the total enzyme solution volume in mL, *d* is the cuvette’s optical path in cm, *ζ* is the absorbance coefficient of 1 M NADH (6.22), Δ*t* is the time from *Eo* to *E*1 in minutes, *Vs* is the total reaction volume in mL and *m* is the sample fresh weight in grams.

### 2.3. RCA Activity Determination

RCA activity was assessed with a Rubisco activase assay kit (Genmed Scientifics Inc., Washington, DC, USA) based on the directions provided. Briefly, 200 mg of the tissue samples was combined with 500 μL of Genmed Lysis Solution (Reagent A), followed by centrifugation for 10 min at 1000 rpm at 4°C. Next, 65 μL of room temperature Genmed buffer (Reagent B) was added to each well of a 96-well plate, followed by the addition of 10 μL of Genmed enzymatic solution (Reagent C) and 10 μL of Genmed reaction solution (Reagent D). The background was evaluated by adding 10 μL of Reagent B to an appropriate well. The reactions were incubated for 3 min at 25 °C, after which they were evaluated with a microplate reader at 0 min and 1 min.

### 2.4. PRK Activity Measurements

PRK activity was measured by diluting aliquots of the extract, and coupling the formation of ADP to the oxidation of NADH with pyruvate kinase and lactate dehydrogenase [34]. For this assay, a solution containing 100 mM Tris-HC1 (pH 7.8), 10 mM MgCl_2_, 20 mM KCI, 0.25 mM NADH, 20 mM DTT, 2 U·mL^−1^ pyruvate kinase, 2 U·mL^−1^ lactate dehydrogenase, 1.5 U·mL^−1^ 1 ribose-5-phosphate (R5P)-isomerase, 1 mM ATP, 0.6 mM phosphoenolpyruvate, 2.5 mM R5P and PRK was used, with NADH oxidation being assessed at 334 nm. Following a 2 min incubation at 25 °C, R5P was added (2.5 mM) to initiate the reaction, and the enzyme activity was determined based upon linear changes in absorbance following the addition of R5 and correcting for the background during the preincubation period.

### 2.5. Assessment of Carbon Assimilation-Related Gene Expression by qRT-PCR

TRIzol (Takara, Dalian, China) and DNase I treatment were utilized to isolate RNA from plant tissues, after which, 1 μg of total RNA per sample was reverse-transcribed into cDNA using reverse transcription enzymes (Takara) based on the directions provided.

Quantitative real-time PCR (qRT-PCR) reactions were conducted using SYBR Premix ExTaq (Takara) and an ABI StepOnePlus Real-Time PCR instrument, with *Df18sRNA* serving as a housekeeping gene [35,36]. Briefly, each 20 μL reaction contained 0.2 μg of cDNA, as well as 10 μL of a 2 × SYBR reaction mix and 0.4 μM of the appropriate primers. The t hermocycler settings were as follows: 95 °C for 30 s, then 40 cycles of 95 °C for 10 s and 60 °C for 30 s. Three biological replicates were independently evaluated for each sample, and the 2^−ΔΔCt^ approach was used to assess the relative gene expression. Melting curve analyses were additionally performed to ensure the reactions’ specificity. The GenBank accession numbers for *DfPRK*, *DfRbcS*, *DfRbcL*, *DfRCA* and *DfRaf* are MW524052, MW524050, MW524049, MW524051 and MW524053, respectively. The primers used in this study are compiled in Table 1.

### 2.6. Subcellular Localization Assay

To generate the subcellular localization assay, the *DfPRK*, *DfRbcS*, *DfRbcL*, *DfRCA* and *DfRaf* coding region was amplified by PCR. The primers used are listed in Table 1. Full-length *DfPRK*, *DfRbcS*, *DfRbcL*, *DfRCA* and *DfRaf* coding sequences were inserted into the pCAMBIA2300-enhanced GFP vector to generate the 35S::*DfPRK*, *DfRbcS*, *DfRbcL*, *DfRCA* and *DfRaf*-EGFP recombinant construct, which was fused with DNA sequences encoding enhanced green fluorescent protein (EGFP). Both 35S::EGFP and the 35S::*DfPRK*, *DfRbcS*, *DfRbcL*, *DfRCA* and *DfRaf*-EGFP constructs were introduced into *Agrobacterium tumefaciens* strain LBA4404. Next, tobacco leaves (*Nicotiana benthamiana*) were injected with *Agrobacterium*-harboring constructs. After 3 days of infiltration with *Agrobacterium tumefaciens*, the EGFP signals were observed under a confocal laser scanning microscope (A1Si, Japan); 4′,6-diamidino-2-phenylindole (DAPI) was used to stain the nucleus.

### 2.7. Generation of Transgenic *Nicotiana tabacum* Plants

A full-length *DfRaf*-coding sequence was inserted into the pCAMBIA2301 vector to generate the 35S::*DfRaf* recombinant construct. The obtained recombinant plasmid 35S::*DfRaf* was introduced into *Agrobacterium tumefaciens* strain LBA4404. Generation of transgenic *Nicotiana tabacum* was performed following the leaf disk method [37]. Transgenic plants were selected using phosphinothricin (5 mg·L^−1^) and confirmed by using PCR. PCR-positive plants were further confirmed by semi-quantitative RT-PCR and quantitative real-time RT-PCR analyses. The primers used are listed in Table 1. We propagated these plants, and then they were used to investigate phenotypes and perform molecular and physiological analyses.

### 2.8. Determination of Physiological Indices

The contents of proline, H_2_O_2_, O_2_^−^, malondialdehyde, soluble sugar and the activities of superoxide dismutase, peroxidase and catalase in tobacco were determined by a proline content kit (PRO-1-Y), an H_2_O_2_ content kit (H_2_O_2_-1-Y), an O_2_^−^ content kit (SA-1-G), a malondialdehyde kit (MDA-1-Y), a plant soluble sugar content kit (KT-1-Y), a superoxide dismutase kit (SOD-1-Y), a peroxidase kit (POD-1-Y) and a catalase kit (CAT-1-W) according to the manufacturer’s instructions (Comin Biotechnology, Suzhou, China).

### 2.9. Histochemical Staining of Hydrogen Peroxide (H_2_O_2_) and Superoxide (O_2_^−^)

Fresh leaves were put into a 1 mg mL^−1^ 3,3′-diaminobenzidine (DAB) solution (pH 5.5, 50 mM Tris-HCl) and a 0.2% nitro blue tetrazolium (NBT) solution (pH 7.8, 50 mM phosphate buffer). The leaf samples were then placed in the dark at 28 °C for 12 h. Finally, decolorization was carried out in a 70 °C boiling water bath with 90% ethanol, and the leaves were stored in 50% glycerol.

### 2.10. Photosynthetic Pigments

Fresh leaf samples were minced, homogenized in a 3 mL solution containing 95% acetone and a limited quantity of CaCl_2_ and quartz sand, then were spun for 30 min at 5000× *g* at 4 °C. Supernatants were then collected, and the levels of chlorophyll a, chlorophyll b and carotenoids therein were measured at absorbance wavelengths of 665, 649 and 470 nm, respectively, using an enzyme-labeled instrument (Thermo Scientific, Waltham, MA, USA). The levels of these three pigments were then quantified as follows: chlorophyll a = 13.95·A665 − 6.88·A649, chlorophyll b = 24.96·A649 − 7.32·A665 and total carotenoids = (1000·A470 − 2.05·Chl a − 114.8·Chl b)/245.

### 2.11. Phylogenetic, Multiple Sequence Alignment and Conservative Motif Analyses

Fragrant woodfern polypeptide sequences similar to Raf (MW524053) were identified using BLAST (E < 1 × 10^−5^) (http://blast.ncbi.nlm.nih.gov (accessed on 1 May 2021)) [38]. The plant sequences used in the phylogenetic analysis were as follows: *Physcomitrium patens* (XP_024402609.1), *Vitis vinifera* (RVW88115.1), *Carex littledalei* (KAF3328238.1), *Jatropha curcas* (XP_012089314.1), *Populus trichocarpa* (XP_002319651.1), *Vigna radiata* (XP_014510931.1), *Cicer arietinum* (XP_004495565.1), *Cajanus cajan* (XP_020203342.1), *Trema orientale* (PON73478.1), *Juglans regia* (XP_018824157.2), *Solanum pennellii* (XP_015058205.1), *Arachis hypogaea* (XP_025655384.1), *Eutrema salsugineum* (XP_006408180.1), *Selaginella moellendorffii* (XP_024536106.1) and *Arabidopsis thaliana* (Q9SR19.1). Alignment was performed using CLUSTALW (http://genome.jp/tools/clustalw (accessed on 1 May 2021)). A maximum likelihood approach was used to construct an unrooted phylogenetic tree based upon amino acid sequences in MEGA 7 (parameter settings: JTT model, γ distribution of the site replacement rate, with 1000 bootstrap reiterations) [39,40]. MEME software (http://meme-suite.org/ (accessed on 1 May 2021)) was additionally used to predict conserved motifs, using the default parameters with the exception of the maximum number of motifs being set to 10 (parameter settings: optimal amino acid residue width, 6–50; any number of repetitions; maximum of 10 more motifs). DNAMAN was used for multiple sequence alignment of each DfRaf.

### 2.12. Statistical Analysis

All experiments were repeated three times, and each biological replicate included three technical replicates. Excel was used for data analysis, and GraphPad Prism 8.0 was used to prepare the figures.

Data were analyzed via one-way analysis of variance (ANOVA). Data were expressed as means or percentages ± standard deviation (SD).

## 3. Results

### 3.1. Carbon Assimilation Gene Expression Patterns in Different Fragrant Woodfern Tissues

The *DfRbcS*, *DfRbcL*, *DfRCA*, *DfRaf* and *DfPRK* genes were expressed in the roots, petioles and leaves of fragrant woodfern plants (Figure 1). The relative expression of *DfRbcS*, *DfRbcL*, *DfRCA* and *DfRaf* in fragrant woodfern leaves was significantly higher than in root and petiole tissues (Figure 1a–d). Relative *DfPRK* levels in fragrant woodfern leaves and petioles were significantly higher than in root tissues (Figure 1e). The relative expression of *DfRbcL*, *DfRCA* and *DfRaf* in fragrant woodfern roots was not significant (Figure 1b–d). Therefore, we selected leaves for our follow-up experimental research.

### 3.2. Effects of High-Temperature Stress on Carbon Assimilation Enzymes’ Activity in Fragrant Woodfern Leaves 

We found that the Rubisco activation state, RCA, PRK activity, initial Rubisco activity and total Rubisco activity first trended upwards and then decreased following exposure to high-temperature stress, reaching peak levels at the 0.5 h time point for the total Rubisco activity, initial Rubisco activity and Rubisco activation state, and at the 1 h time point for RCA and PRK activities. In comparison with the 0 h time point, the total Rubisco activity, initial Rubisco activity and Rubisco activation state were significantly increased at 0.5 h, but significantly decreased from 1 to 48 h (Figure 2a–c). RCA and PRK activities significantly increased from 0.5 to 1 h relative to 0 h, but significantly decreased from 3 to 48 h (Figure 2d,e).

### 3.3. Effects of High-Temperature Stress on the Relative Expression of Carbon Assimilation Enzymes Genes in Fragrant Woodfern Leaves 

Under high-temperature stress, the relative expression of *DfRbcS*, *DfRbcL*, *DfRCA* and *DfPRK* in fragrant woodfern leaves first increased and then decreased over time, reaching peak levels at the 0.5 h time point. The relative expression of *DfRbcS* and *DfRbcL* in fragrant woodfern leaves was significantly higher at 0.5 h relative to 0 h, whereas these levels were significantly lower at the 1 h and 6–48 h time points (Figure 3a,b). Relative to 0 h, the relative expression of *DfRCA* in fragrant woodfern leaves was significantly increased at 0.5–3 h but was significantly decreased at 6–48 h (Figure 3c). The relative expression of *Dfraf* in leaves of fragrant woodfern was significantly higher than 0 h at 0.5–24 h (Figure 3d). The relative expression of *DfPRK* in leaves of fragrant woodfern was significantly higher than 0 h at 0.5 h but significantly lower than 0 h at 3–48 h (Figure 3e). According to this qRT-PCR analysis, the high-temperature treatment upregulated the expression of *DfRaf*, which was significantly higher than that of the other genes. Therefore, *DfRaf* was selected for further study.

### 3.4. Subcellular Location of Carbon Assimilation Enzyme Genes

A subcellular localization experiment with 35S::*DfRbcS*, *DfRbcL*, *DfRCA*, *DfRaf* and *DfPRK*-GFP was performed in tobacco leaves to explore the potential function of the *DfRbcS*, *DfRbcL*, *DfRCA*, *DfRaf* and *DfPRK* genes. The GFP fluorescence of the 35S::*DfRbcS*, *DfRbcL*, *DfRCA* and *DfPRK*-GFP fusion protein was detected in the chloroplast. The GFP fluorescence of the 35S::*DfRaf* fusion protein was detected in the nucleus (Figure 4). 

### 3.5. Phylogenetic, Multiple Sequence Alignment and Conservative Motif Analysis of Fragrant Woodfern Raf Proteins

We next constructed a phylogenetic tree based upon the Raf protein sequences from 18 different plant species, including the monocotyledonous *Carex littledalei*, the dicotyledonous *Arabidopsis thaliana*, the bryophyte *Physcomitrella patens* and the pteridophyte *Selaginella moellendorffii*. This analysis revealed DfRaf to be closely related to SmRaf and PpRaf and less closely related to angiosperm Raf proteins (Figure 5). Conservative motif analysis revealed that the lengths of the 10 identified motifs ranged from 14 to 50 amino acids. Motif 1 included the Raf1_HTH domain, while Motifs 2–3 included the Raf1_N domain (Figure 5), and Motifs 4, 6, 7 and 8 included the RuBisCo_chap_C domain. Motifs 1–3 and 5–9 were present in all Raf protein sequences (Figure 5). Conserved domain analyses of these Raf proteins were conducted using NCBI-CDD (https://www.ncbi.nlm.nih.gov/Structure/cdd/wrpsb.cgi (accessed on 1 May 2022)), revealing that all Raf sequences to contain Raf1_N, RuBisCo_chap_C and Raf1_HTH (Table 2). In total, three plant Raf protein sequences (*A. thaliana*, *P. patens* and *S. moellendorffii*) were selected for alignment using DNAMAN to evaluate relatedness among these proteins, revealing highly conserved sequences corresponding to the Raf1_N, RuBisCo_chap_C and Raf1_HTH domains (Figure 6).

### 3.6. Overexpression of DfRaf Increased the High-Temperature Tolerance of Transgenic Tobacco

To describe the function of *DfRaf* in tolerance against heat stress, we obtained six independent *Nicotiana tabacum* lines overexpressing *DfRaf* (DfRaf-OV-L1-6). The *DfRaf* gene in these lines was detected by genomic DNA PCR, and the results showed that lines DfRaf-OV-L1-6 were PCR-positive. Semi-quantitative reverse transcription (RT)-PCR showed that the levels of *DfRaf* transcripts in DfRaf-OV-L2 and DfRaf-OV-L6 were significantly higher. Therefore, DfRaf-OV-L2 and DfRaf-OV-L6 were chosen for phenotypic analysis. Under normal conditions, there was no significant difference in phenotypes between the DfRaf-OV-L2, DfRaf-OV-L6, VC and WT plants. Compared with the leaves of DfRaf-OV-L2 and DfRaf-OV-L6, the leaves of WT and VC turned yellow under the high-temperature treatment (Figure 7).

Photosynthetic pigments are an important place for green plants to carry out photosynthesis. The increase in photosynthetic pigment content is conducive to the accumulation of more photosynthetic products in plants and improves the ability of plants to resist stress [41]. The chlorophyll a, chlorophyll b and total carotenoid content of DfRaf-OV-L2 and DfRaf-OV-L6 were significantly higher than those of WT and VC under the heat treatment (Figure 8a–c). Soluble sugars are important osmotic regulators and nutrients. Their increase and accumulation protect biofilms [42,43,44,45]. In comparison with those of the WT and VC plants, the soluble sugar content of DfRaf-OV-L2 and DfRaf-OV-L6 significantly increased under the heat treatment (Figure 8d). In addition, plants also produce a large amount of proline to cope with abiotic stress. Proline accumulated in plants can be used as an osmotic regulator to regulate the intracellular permeability of cytoplasm. It also stabilizes the structure of biological macromolecules, reduces the acidity of cells and relieves ammonia toxicity [46]. The proline content of DfRaf-OV-L2 and DfRaf-OV-L6 was significantly higher than that of WT and VC under heat treatment (Figure 8e). Malondialdehyde (MDA) is the main product of peroxidative degradation of membrane lipids, and it is also an important indicator for measuring the degree of membrane lipid peroxidation [41,47]. In comparison with the WT and VC, the MDA content of DfRaf-OV-L2 and DfRaf-OV-L6 was significantly decreased under heat treatment (Figure 8f). Under abiotic stress, ROS will accumulate rapidly in plants, and high concentrations of ROS will directly affect the normal development of cells [48]. To maintain intracellular ROS balance, plants have evolved a series of enzymatic and non-enzymatic ROS-scavenging mechanisms. The H_2_O_2_ and O_2_^−^ content of DfRaf-OV-L2 and DfRaf-OV-L6 was significantly lower than that of WT and VC under heat treatment (Figure 8g–h and Figure 9). The POD, SOD and CAT levels of DfRaf-OV-L2 and DfRaf-OV-L6 were significantly higher than those of WT and VC under heat treatment (Figure 8i–k).

### 3.7. Effects of High-Temperature Stress on Carbon Assimilation Enzyme Activity in Transgenic Tobacco

High-temperature stress can reduce the efficiency of carbon assimilation in plants. In normal circumstances, compared with the WT and VC, the Rubisco activation state, RCA, PRK activity, initial Rubisco activity and total Rubisco activity of DfRaf-OV-L2 and DfRaf-OV-L6 were significantly increased (Figure 10). The Rubisco activation state, RCA, PRK activity, initial Rubisco activity and total Rubisco activity of DfRaf-OV-L2 and DfRaf-OV-L6 were significantly higher than those of WT and VC under heat treatment (Figure 10).

## 4. Discussion

High-temperature stress can affect the photosynthetic capacity of plants by inhibiting photosynthetic electron transport and photosynthetic phosphorylation processes [49]. Rubisco is the central Calvin cycle carboxylase in C3 plants [19,50], and is composed of 16 subunits (eight RbcL and eight RbcS subunits) [51]. RbcL is encoded by the chloroplast genome and contains an active carboxylation/oxygenation catalytic site [52], while RbcS is encoded by a nuclear gene and serves to modulate the catalytic activity of this complex [53]. We found that *DfRbcL* and *DfRbcS* expression levels in fragrant woodfern leaves were rapidly responsive to high-temperature stress. As the duration of the high-temperature treatment increased, the relative expression of *DfRbcL* and *DfRbcS* decreased. The synthesis of Rubisco, a key photosynthetic enzyme, was thus hindered, adversely impacting photosynthesis, plant growth and development. RbcL has previously been shown to be involved in adaptation to low-temperature stress conditions, whereas RbcS is more important for maintaining Rubsico activity under optimal conditions [54,55]. We found Rubisco activity to be inconsistent with the expression of *DfRbcL* and *DfRbcS*, suggesting that temperature stress may have disrupted the synthesis of the Rubisco enzymatic complex, thus affecting photosynthesis. Rubisco can make up roughly half of the soluble sugar found in plant leaves, but it exhibits poor catalytic efficiency [56]. The mechanisms regulating Rubisco activity are dependent upon the AAA+ (ATPases associated with a variety of cellular activities) family protein RCA, which exhibits ATP hydrolase activity [57], utilizing the energy of ATP hydrolysis to release the inhibitors bound to Rubisco, change the conformation of Rubisco and activate it [58]. Under high-temperature stress, RCA expression can be downregulated, impairing Rubisco activation and the photosynthesis rate [7,59]. We observed that *DfRCA* expression levels in fragrant woodfern leaves responded to high-temperature stress within 0.5–3 h. As the duration of the high-temperature treatment increased, the relative expression of *DfRCA* decreased, with RCA activity exhibiting a comparable trend. Raf plays a role in releasing and/or isolating RbcL in chaperone proteins early during the Rubisco assembly process. Rubisco assembly is thus chaperone-assisted, and the folding of newly synthesized RbcL requires the chaperonin proteins GroEL and GroES [51]; the subsequent assembly of folded RbcL monomers into oligomers is triggered by binding with the Raf chaperones [18]. Although Raf1 is evolutionarily conserved, its importance in plants and cyanobacteria remains controversial. In maize, genetic deletion of Raf1 resulted in the depletion of Rubisco and plant lethality [11], while the overexpression of Rubisco subunits in combination with Raf1 was able to increase Rubisco protein content and activity [60]. In addition, *Arabidopsis* Raf1 can promote the assembly of recombinant Rubisco in tobacco chloroplasts [61]. The exact roles of Raf1 in cyanobacteria remain enigmatic [19]. In vitro reconstitution experiments have implied that Raf1 functions in stabilizing the antiparallel RbcL2 dimer [62]. In this study, DfRaf exhibited conserved Raf1_HTH, Raf1_N and RuBisCo_chap_C domains. Raf1 from *Arabidopsis thaliana* consists of an N-terminal α-domain, a flexible linker segment and a C-terminal β-sheet domain that mediates dimerization. The α-domains mediate the majority of functionally important contacts with RbcL by bracketing each RbcL dimer at the top and bottom of this complex [18]. The α-domain alone is essentially inactive [63]. The gene expression patterns of *DfRaf* exhibited highly complex characteristics in response to high-temperature stress, but the mechanisms of the response of *DfRaf* to these conditions remain unclear and require further exploration. Rubisco activity is not only regulated by Raf and RCA, but it also interacts with various phosphate sugars and loses activity. For example, the substrate RuBP is itself a strong inhibitor of Rubisco. PRK catalyzes the regeneration of RuBP and provides a receptor for the assimilation of CO_2_. The results of this study indicated that high-temperature stress significantly inhibited the expression of *DfPRK* and PRK enzyme activity in fragrant woodfern leaves. Studies have shown that plant *PRK*, *RbcL*, *RbcS*, *RCA* and *Raf* genes exhibit tissue-specific expression patterns. Liu et al. found that *AtPRK*, *AtRbcL*, *AtRbcS*, *AtRCA* and *AtRaf* genes are only expressed in green tissues such as the leaves, stems and siliques of *Arabidopsis thaliana* [64]. Watillon et al. found that the *MaPRK*, *MaRbcL*, *MaRbcS*, *MaRCA* and *MaRaf* genes in *Malus domestica* can also be detected in large quantities in the leaves, but they are almost undetectable in roots and stems [65]. We determined that the relative *DfRbcS*, *DfRbcL*, *DfRCA* and *DfRaf* expression levels in fragrant woodfern leaves were significantly higher than those in root or petiole tissues, consistent with prior findings [64,65].

High temperature can cause the production of ROS in plants. Low or moderate concentrations of ROS can be used as signaling molecules, and high concentrations of ROS are harmful to plants [66]. When the concentration of ROS is high, plants will activate their own antioxidant systems to alleviate the oxidative damage caused by ROS, including the antioxidant enzymes SOD, CAT and POD [67]. The results of this study also confirmed that DfRaf-OV-L2 and DfRaf-OV-L6 enhanced the activity of antioxidant enzymes, reduced the release of ROS and effectively alleviated the damage to plants caused by high temperature. With further warming of the global climate, high temperature may seriously affect the photosynthesis of plants [68]. The photosynthesis and growth of plants are usually limited by the activity of the carbon dioxide fixer Rubisco [69], and Raf is a key mediator regulating the activity of Rubisco [70]. Raf1 was the first factor characterized as an assembly chaperone involved in Rubisco biogenesis in the chloroplasts [13]. Improving the expression of Raf1 can improve the performance of Rubisco, which is a feasible way to significantly improve plant yield [18]. In this study, the Rubisco activation state, RCA, PRK activity, initial Rubisco activity and total Rubisco activity of DfRaf-OV-L2 and DfRaf-OV-L6 were significantly higher than those in the WT and VC under heat treatment. We speculate that Raf can promote Rubisco activity, and thereby bolsters carbon fixation efficiency and plants’ carbohydrate content. Carbohydrates replace the water bound to the membrane structures of plants and biological macromolecules under conditions of adversity such as high-temperature stress, thereby protecting and stabilizing biological macromolecules. The related mechanisms of action warrant further study.

## 5. Conclusions

We found that the high-temperature treatment upregulated the expression of *DfRaf*. Overexpression of *DfRaf* led to the increased activity of Rubisco, RCA and PRK under high-temperature stress. The photosynthetic pigments, proline, soluble sugar content and ROS-scavenging ability of the DfRaf-OV-L2 and DfRaf-OV-L6 transgenic lines were higher than those of WT and VC plants under high-temperature stress. The results showed that *DfRaf* could regulate the carbon assimilation rate to enhance the high-temperature tolerance of plants by increasing Rubisco activity.

## Figures and Tables

**Figure 1 genes-13-01212-f001:**
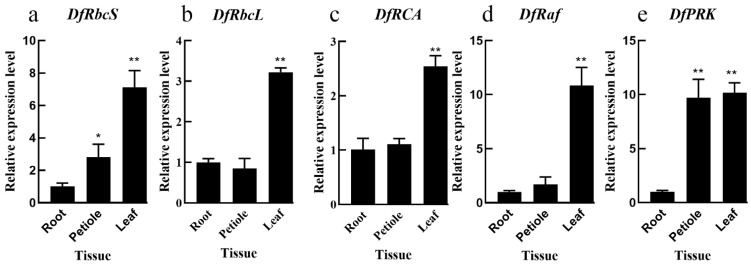
The relative expression levels of *DfRbcS* (**a**), *DfRbcL* (**b**), *DfRCA* (**c**), *DfRaf* (**d**) and *DfPRK* (**e**) in different fragrant woodfern tissues. Each bar corresponds to the mean value  ±  SD of three independent replicates. * *p* < 0.05; ** *p* < 0.01.

**Figure 2 genes-13-01212-f002:**
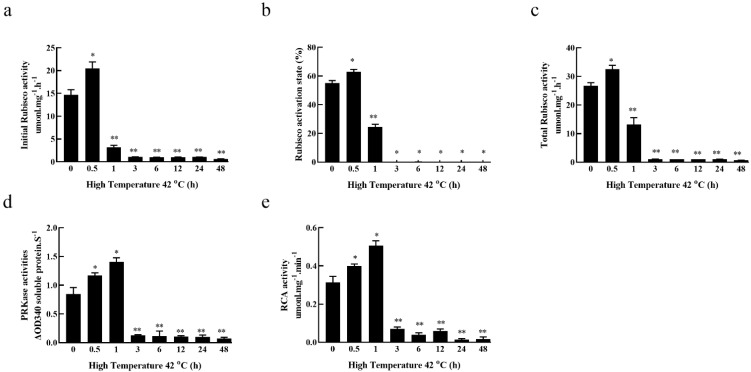
Effects of high-temperature stress on carbon assimilation enzyme activity in the leaves of fragrant woodfern at different times. Effects of high-temperature exposure on total Rubisco activity (**a**), initial Rubisco activity (**b**), Rubisco activation state (**c**), RCA activity (**d**) and PRK activity (**e**) in the leaves of fragrant woodfern at different times. Each bar in the graph corresponds to the mean value  ±  SD of three independent replicates. * *p* < 0.05; ** *p* < 0.01.

**Figure 3 genes-13-01212-f003:**
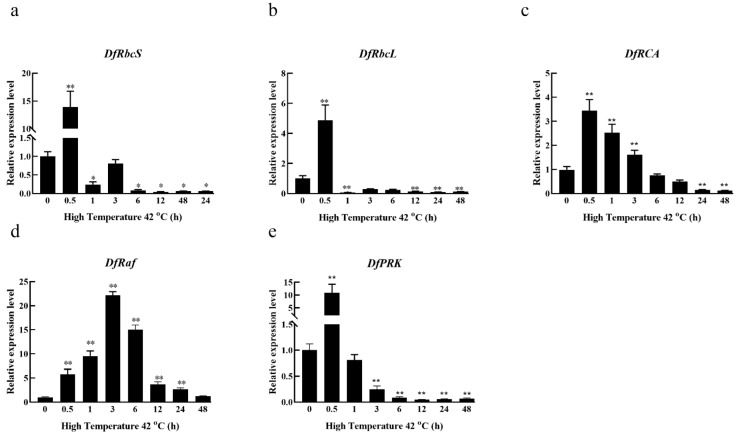
Effects of high-temperature stress on the relative expression of carbon assimilation enzyme genes in fragrant woodfern at different times. Effects of high-temperature stress on the relative expression level of *DfRbcS* (**a**), *DfRbcL* (**b**), *DfRCA* (**c**), *DfRaf* (**d**) and *DfPRK* (**e**) in the leaves of fragrant woodfern at different times. Each bar corresponds to the mean value  ±  SD of three independent replicates. * *p* < 0.05; ** *p* < 0.01.

**Figure 4 genes-13-01212-f004:**
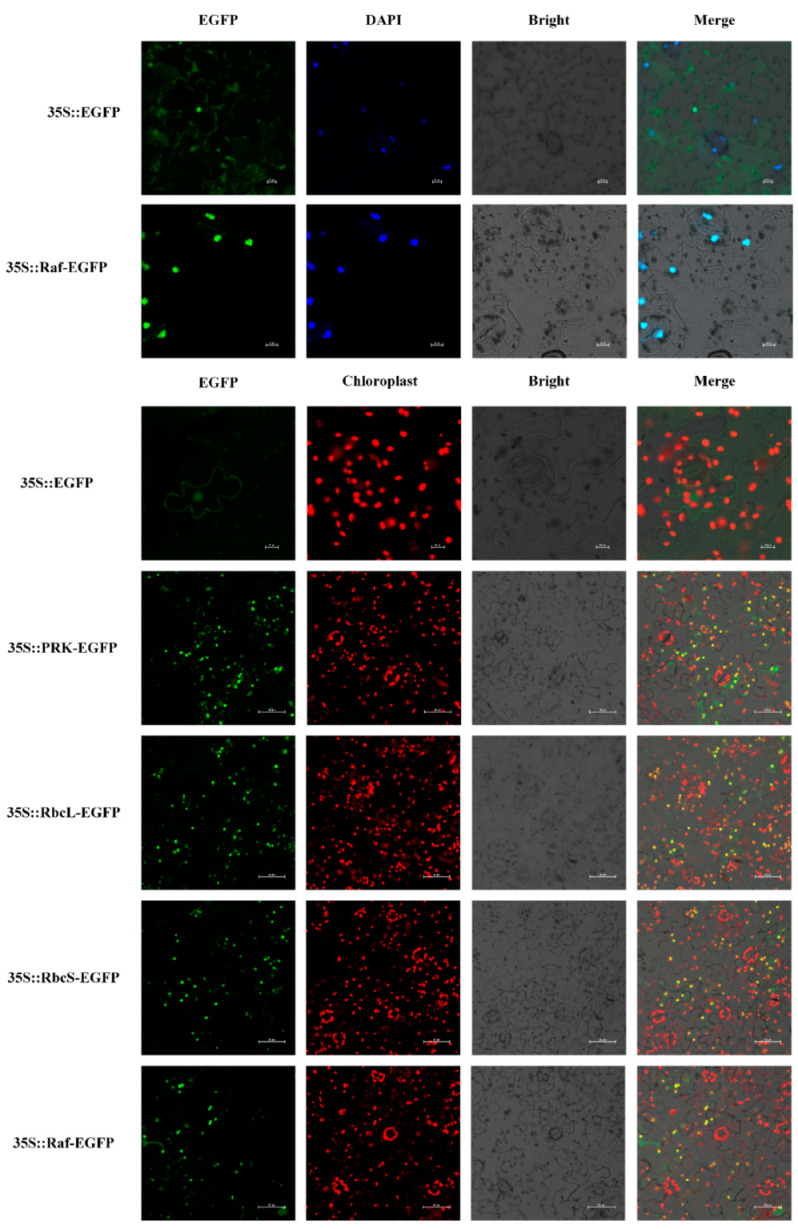
Subcellular localization of DfRbcS, DfRbcL, DfRCA, DfRaf and DfPRK proteins in tobacco leaf cells. The red fluorescence of DAPI indicates the nucleus; bar = 10 μm.

**Figure 5 genes-13-01212-f005:**
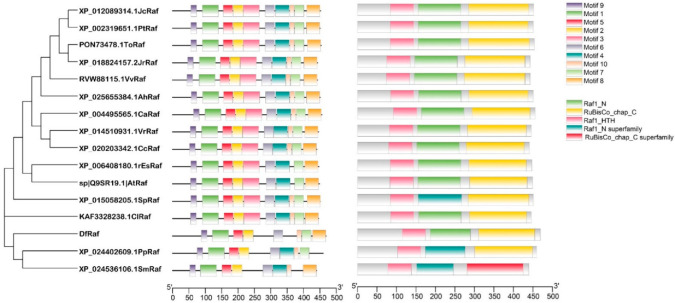
Phylogenetic tree and conserved motif analyses for the DfRaf protein. A maximum likelihood approach was used to construct a phylogenetic tree based upon amino acid sequences using MEGA 7 (parameter settings: JTT model, γ distributions of the site replacement rate, with 1000 bootstrap replicate iterations).

**Figure 6 genes-13-01212-f006:**
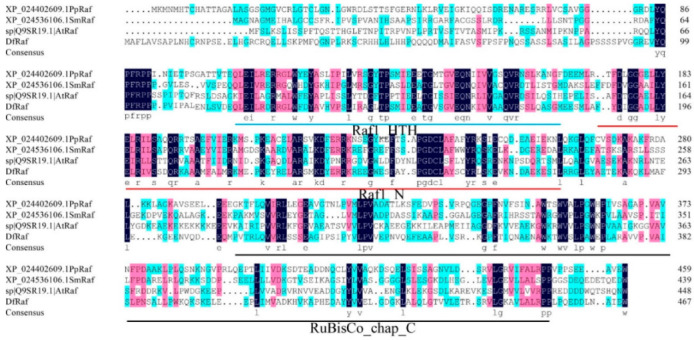
Multiple sequence alignment for the DfRaf protein.

**Figure 7 genes-13-01212-f007:**
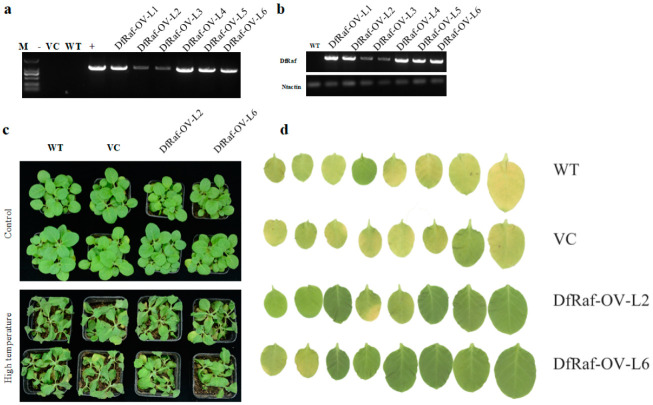
Genomic DNA PCR and semi-quantitative reverse transcription (RT)-PCR identification of transformation with *DfRaf* and the phenotype of tobacco transformed with the *DfRaf* gene after heat treatment. (**a**) PCR identification of transformation with *DfRaf*. M: Trans2K DNA marker; -: ddH_2_O negative control; +: positive control; VC: vector control; WT: wild-type; *DfRaf* overexpression lines: DfRaf-OV-L1-6. (**b**) Semi-quantitative reverse transcription (RT)-PCR identification of transformation with *DfRaf*. WT: wild-type; DfRaf-OV-L1-6: transformation with the *DfRaf* gene. (**c**) Phenotype of tobacco transformed with the *DfRaf* gene under heat treatment. WT: wild-type; DfRaf-OV-L2 and DfRaf-OV-L6 were transformed with the *DfRaf* gene. (**d**) Phenotype of the leaves of WT, VC, DfRaf-OV-L2 and DfRaf-OV-L6 under the high-temperature treatment.

**Figure 8 genes-13-01212-f008:**
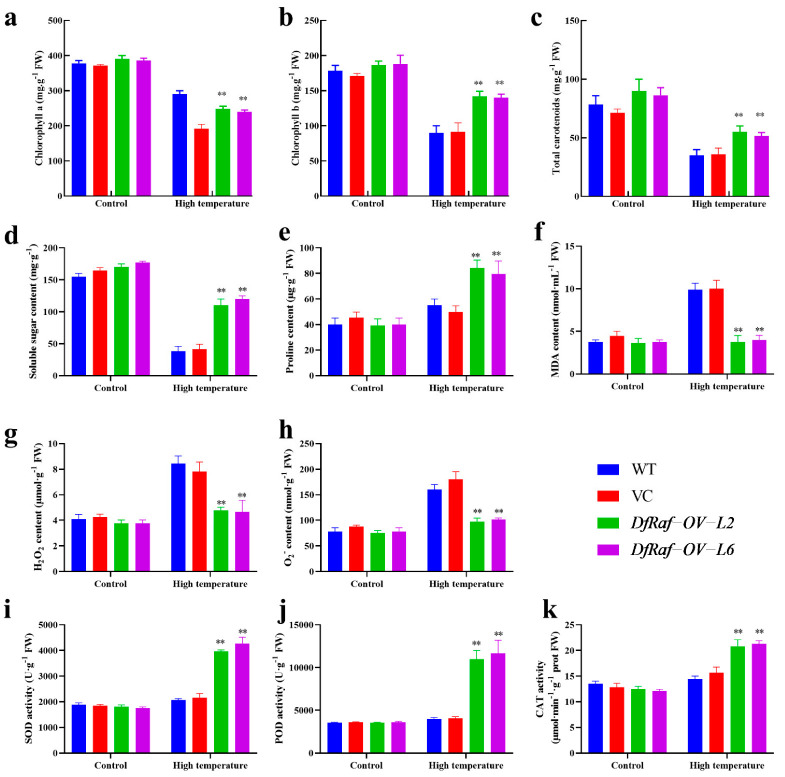
Physiological indices of *DfRaf* overexpression lines under heat stress. The content of chlorophyll a (**a**), chlorophyll b (**b**), total carotenoids (**c**), soluble sugar (**d**), proline (**e**), MDA (**f**), H_2_O_2_ (**g**), O_2_^−^ (**h**) and the activity of SOD (**i**), POD (**j**) and CAT (**k**) of tobacco lines under heat stress; Each bar in the graph corresponds to the mean value  ±  SD of three independent replicates. ** *p* < 0.01.

**Figure 9 genes-13-01212-f009:**
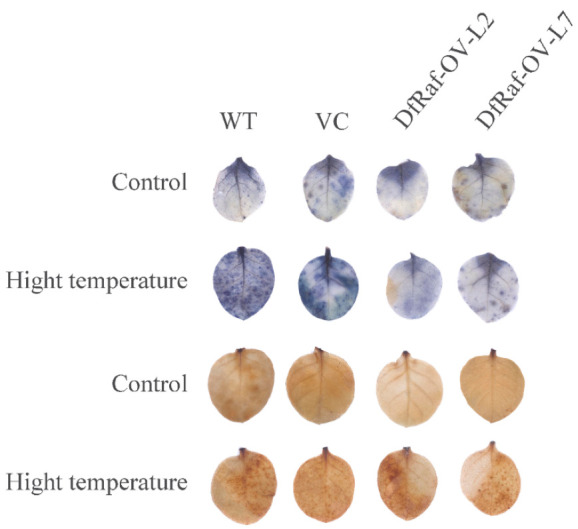
DAB and NBT staining of tolerant leaves under high-temperature stress.

**Figure 10 genes-13-01212-f010:**
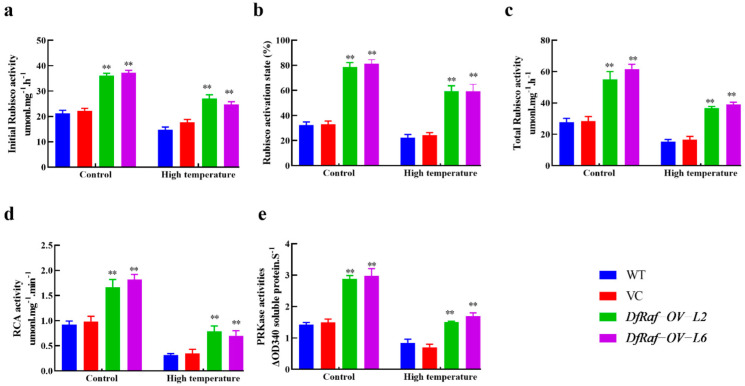
Effects of high-temperature stress on carbon assimilation enzyme activity in the leaves of *DfRaf* overexpression lines. Effects of high-temperature exposure on total Rubisco activity (**a**), initial Rubisco activity (**b**), Rubisco activation state (**c**), RCA activity (**d**) and PRK activity (**e**) in the leaves of *DfRaf* overexpression lines. Each bar in the graph corresponds to the mean value  ±  SD of three independent replicates. ** *p* < 0.01.

**Table 1 genes-13-01212-t001:** Primers for carbon assimilation-related genes.

Gene ID	GenBank	Forward Primer (5′ to 3′)	Reverse Primer (5′ to 3′)	Purpose
*DfRbcL*	MW524049	GGAGACTAAAGCAGGTGTTG	GTCAACCCATCTGTCCATAC	qRT-PCR
*DfRbcS*	MW524050	ACTGGTAAGGACAGGTGGAA	CCATGAAAGCAGGAGCAA	qRT-PCR
*DfRCA*	MW524051	CGTCCACTGCCACTACTCCT	TCGTAAGCAAGGCCACCC	qRT-PCR
*DfPRK*	MW524052	CTGCCCTCCCTTCTTCGT	GCAGCACCTCCGAAGACA	qRT-PCR
*DfRaf*	MW524053	TGGGCTGTGGTTTGACTA	AGGCGAGCATACTTTCTTC	qRT-PCR
Df18sRNA		GCTTTCGCAGTAGTTCGTCTTTC	TGGTCCTATTATGTTGGTCTTCGG	qRT-PCR
*DfRbcL*		ATGTCACCACAAACGGAGACT	TTACAACGTATCAATTGTCTCG	Gene cloning
*DfRbcS*		ATGGCTACTACTGTAGCTGCT	CTCCCGCACTCATCATGATTG	Gene cloning
*DfRCA*		ATGGCGTCCACTGCCACTACTC	TCACTTGGGGTCTGCAAAATCA	Gene cloning
*DfPRK*		ATGAAACGCATTCACATATTC	TCAAGCCTTAGTGGATTGCAAA	Gene cloning
*DfRaf*		ATGGCGTTTTTGGCGGTCTCA	TTAGTCCCATTCTATAGCATTC	Gene cloning
*DfRbcL*-EGFP		CTCGGTACCCGGGGATCC ATGTCACCACAAACGGAGACT	GGTGTCGACTCTAGAGGATCC TTACAACGTATCAATTGTCTCG	Gene cloning
*DfRbcS*-EGFP		CTCGGTACCCGGGGATCC ATGGCTACTACTGTAGCTGCT	GGTGTCGACTCTAGAGGATCC CTCCCGCACTCATCATGATTG	Gene cloning
*DfRCA*-EGFP		CTCGGTACCCGGGGATCC ATGGCGTCCACTGCCACTACTC	GGTGTCGACTCTAGAGGATCC TCACTTGGGGTCTGCAAAATCA	Gene cloning
*DfPRK*-EGFP		CTCGGTACCCGGGGATCC ATGAAACGCATTCACATATTC	GGTGTCGACTCTAGAGGATCC TCAAGCCTTAGTGGATTGCAAA	Gene cloning
*DfRaf*-EGFP		CTCGGTACCCGGGGATCC ATGGCGTTTTTGGCGGTCTCA	GGTGTCGACTCTAGAGGATCC TTAGTCCCATTCTATAGCATTC	Gene cloning
*pCAMBIA2301-DfRaf*		GGGCATCGATACGGGATCCAT ATGGCGTTTTTGGCGGTCTCA	TCGAGCTCGATGGATCCCGTA TTAGTCCCATTCTATAGCATTC	Gene cloning

**Table 2 genes-13-01212-t002:** Motif sequences.

No.	Sequence (5′ to 3′)	Number	Pfam
Motif1	ILANRLGLWYEYAPLIPSLIREGFTPPSIEEATGISGVEQNRLVVAAQVR	50	Raf1_HTH
Motif2	NRFDPKGAQDLARAMKDFPRRRGDKGWESFD	31	Raf1_N
Motif3	YTLPGDCLSFMYYRQSREHKNPSEQRTAALEQALEVAETEKAKNRILEEL	50	Raf1_N
Motif4	KEILEAPWECRSEGEFGVVVAEKGWKRWVVLPGWEPVVGLGKG	43	RuBisCo_chap_C
Motif5	AFDTGGAELLYEIRLLSASQRAAAARFIV	29	-
Motif6	VRVPVVRLKIGEVAEASSVVVLPVCKAEE	29	RuBisCo_chap_C
Motif7	NRWYKEEPILVVADRGRKEVEADDGFYLV	29	RuBisCo_chap_C
Motif8	GLKVERGSALKERGVKESLGTVVLVVRPPREETDDQLSDED	41	RuBisCo_chap_C
Motif9	PSPPPPQQQLYQPFRPPP	18	-
Motif10	VVVSFPDARVLPWK	14	-

## Data Availability

Not applicable.

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
