# Peer review of "Overexpression of DfRaf from Fragrant Woodfern (Dryopteris fragrans) Enhances High-Temperature Tolerance in Tobacco (Nicotiana tabacum)"

_genes, 2022, doi:10.3390/genes13071212_

Round 1

Reviewer 1 Report

The authors have shown the role of DfRAF1 in heat stress. It is a good study however I feel several lacunae throughout the manuscript about the rationale of an experiment conducted by the author. Authors should read their manuscript thoroughly. There are a lot of typographical, and grammatical mistakes. I strongly feel language shouldn’t be a barrier to understanding science. I feel to understand science we don’t need to know Shakespeare’s English but the language written in the manuscript is hard and not engaging for the reader. Authors can take help from a person/company whose writing skills are good.

Experiment: Authors can conduct DAB and NBT in overexpressed lines. The proline content can also be performed by authors to show the tolerance of overexpressed lines.

If authors have the facility of LICOR in their institute, they can also perform that. It will strongly support the manuscript.  Chlorophyll a, b, carotenoids, and secondary metabolites such as phenolics, flavonoids, and anthocyanins in overexpressed lines under heat stress.

Insilico analysis of DfRAF1 can be improved.

Abstract:

The first line should be removed.

Introduction:

Line26: Both should be removed.

Line37-40: sentence construction is wrong

Line82-89: these lines are repeated twice.

Materials and Methods

How authors have collected the tissue from the control? Did the authors have collected at 0 hr and considered it as a control? or Did The authors collect at all the timepoint and pooled in for control tissue?

What is the rationale of heat treatment? Why do plants directly shift to 42°C? provide the reference of the sudden temperature? Is 42°C too high for nicotiana?

Line 105: forfive times

Line 107: plantletsfilled; These kind of merged words are throughout in the manuscript.

Line 96: Fragrant Woodfern (Dryopteris fragrans) seedlings were propagated from the laboratory of plant resources and molecular biology of Northeast Agricultural University. It should be procured instead of propagated.

Line 135-139: It shouldn’t be a part of materials and methods

Results:

Rubisco large and small subunits have more than 1 copy in the genome. Did the authors try to look at the expression of other copies as well?

Section 3.2, one sentence has written in 6 lines with so many conjunctions and prepositions which is improper English and hard to follow for the reader.

Fig 2 b shows the rubisco activation state in low-temperature conditions. What is the relevance of this?

Fig 2 and 3, why authors did the Student’s t-test here? ANOVA will be appropriate for the data presented here.

Fig 4 Authors have written DfPbcS, DfPbcL instead of DfRbcS, DfRbcL

Line278: “DfRaf was selected for the further study” instead of “DfRaf was selected for the next study”.

Line294: 35S:: DfRaf space should be omitted.

Section 3.5: There is no clear-cut information on how authors have got Raf proteins from other species. How many RAF copies are in Fragrant Woodfern? The authors should conduct a genome-wide analysis for RAF.

The quality of the phylogram is very poor. Authors can consider the following paper for phylogram and genome-wide analysis. The figure needs to be reconstructed

Tiwari, M., Yadav, M., Singh, B., Pandey, V., Nawaz, K. and Bhatia, S. (2021). Evolutionary and functional analysis reveals CaRR13, a TypeB RR as positive regulator of symbiosis. Plant Biotechnol. J.

Tiwari, M., Pandey, V., Singh, B., Yadav, M. and Bhatia, S. (2021) Evolutionary and expression dynamics of LRR-RLKs and functional establishment of KLAVIER homolog in shoot mediated regulation of AON in chickpea symbiosis. Genomics

Pareek A., Mishra, D., Shekhar, S., Chakraborty, S., & Chakraborty, N. (2021). The small heat shock proteins, chaperonin 10, in plants: An evolutionary view and emerging functional diversity. Environmental and Experimental Botany, 182, 0432

Mishra, D., Shekhar, S., Chakraborty, S. and Chakraborty, N. (2018) Carboxylate clamp tetratricopeptide repeat (TPR) domain containing Hsp90 cochaperones in Triticeace: an insight into structural and functional diversification. Environ. Exp. Bot. 155, 31– 44

Mishra D., Suri G.S., Kaur G., Tiwari M. (2021): Comprehensive analysis of structural, functional, and evolutionary dynamics of Leucine Rich Repeats-RLKs in Thinopyrum elongatum. International Journal of Biological Macromolecules, 183: 513–527. https://doi.org/10.1016/j.ijbiomac.2021.04.137

There is no reference for this “soluble sugar are an important osmotic regulator and nutrient. Its increase and accumulation protect biofilms” why the authors have chosen to do soluble sugar estimation in stress. Is there is any reference regarding that? Previously few studies have been performed to estimate soluble sugar in stress and developmental stages such as

Shekhar, S., Mishra, D., Buragohain, A. K., Chakraborty, S., & Chakraborty, N. (2015). Comparative analysis of phytochemicals and nutrient availability in two contrasting cultivars of sweet potato (Ipomoea batatas L.). Food Chemistry, 173, 957–965.

Mishra, D., Shekhar, S., Chakraborty, S., Chakraborty, N. (2021). Wheat 2-Cys peroxiredoxin plays a dual role in chlorophyll biosynthesis and adaptation to high temperatures. Plant J. 105,1374-1389

There are no references for MDA and chlorophyll and no clear rationale why did authors perform all these assays. There are a lot of physiological studies conducted on many crops under heat stress. Authors can refer to them and make the connection in their manuscript.

Pareek A., Rathi D., Mishra, D., Shekhar, S., Chakraborty, S., & Chakraborty, N. (2019). Physiological plasticity to high temperature stress in chickpea: Adaptive responses and variable tolerance. Plant Sciences, 289, 110258.

Mishra, D., Shekhar S., Chakraborty S., Chakraborty N. (2017). Cultivar-specific high temperature stress responses in bread wheat (Triticum aestivum L.) associated with physicochemical traits and defense pathways. Food Chemistry, 221, 1077-1087

Fig 7 a. where is the control for genomic DNA?

Fig 7 b. Why there is no marker in the picture? How much similarity between the DfRAF and NtRAF sequences? The authors have made primer for RT-PCR exclusively for DfRAF or is there some similarity between DfRAF and NtRAF sequences which is chosen for RT-PCR primers.

Line 406: Herein; These are two separate words ‘Here’ and ‘in’. Please rectify throughout the manuscript.

 Line470: con-ditions

Author Response

Response to Reviewer 1 Comments

Point 1:

The authors have shown the role of DfRAF1 in heat stress. It is a good study however I feel several lacunae throughout the manuscript about the rationale of an experiment conducted by the author. Authors should read their manuscript thoroughly. There are a lot of typographical, and grammatical mistakes. I strongly feel language shouldn’t be a barrier to understanding science. I feel to understand science we don’t need to know Shakespeare’s English but the language written in the manuscript is hard and not engaging for the reader. Authors can take help from a person/company whose writing skills are good.

Response 1:

Thank you very much for your question and help us to improve the quality of our manuscript.

The article has been polished by the polishing company.

Point 2:

Experiment: Authors can conduct DAB and NBT in overexpressed lines. The proline content can also be performed by authors to show the tolerance of overexpressed lines.

Response 2:

Thank you very much for your question and help us to improve the quality of our manuscript.

conduct DAB, NBT and proline has been detected.

Plants also produce a large amount of proline to cope with abiotic stress. Proline accumulated in plants can be used as an osmotic regulator to regulate the intracellular permeability of cytoplasm. It also stabilizes the structure of biological macromolecules, reduces the acidity of cells and relieves ammonia toxicity. The proline content of DfRaf-OV-L2 and DfRaf-OV-L6 were significantly higher than it in WT and VC under heat treatment (Fig. 8e; P < 0.01). Undering abiotic stress, ROS will accumulate rapidly in plant and high concentrations of ROS will directly affect the normal development of cells. To maintain intracellular ROS balance, plants have evolved a series of enzymatic and non-enzymatic ROS scavenging mechanisms. H2O2 and O2.− content of DfRaf-OV-L2 and DfRaf-OV-L6 were significantly lower than it in WT and VC under heat treatment (Fig. 8g and h, Fig. 9; P < 0.01);

Fig. 8 Physiological indices of DfRaf overexpression lines under heat stress. The content of chlorophyll a (a), chlorophyll b (b), total carotenoids (c), soluble sugar (d), proline(e), MDA (f), H2O2 (g), O2- (h) and activity of SOD (i), POD (j), SOD (k) of tobacco lines under heat stress; Each bar in the graph corresponds to the mean value ± SD of three independent replicates. * P < 0.05; ** P < 0.01.

Fig. 9 DAB and NBT staining of tolerance leaves under high temperature stress.

Point 3:

If authors have the facility of LICOR in their institute, they can also perform that. It will strongly support the manuscript. Chlorophyll a, b, carotenoids, and secondary metabolites such as phenolics, flavonoids, and anthocyanins in overexpressed lines under heat stress.

Response 3:

Chlorophyll a, b, carotenoids has been detected. Further research on secondary metabolites as follow-up subjects.

Photosynthetic pigments are an important place for green plants to carry out photosynthesis. Photosynthetic pigments are an important place for green plants to carry out photosynthesis. The increase of photosynthetic pigments content is conducive to the accumulation of more photosynthetic products in plants and improves the ability of plants to resist stress. Chlorophyll a, Chlorophyll b and total carotenoids content of DfRaf-OV-L2 and DfRaf-OV-L6 were significantly higher than it in WT and VC under heat treatment (Fig. 8a, b and c; P < 0.01).

Fig. 8 Physiological indices of DfRaf overexpression lines under heat stress. The content of chlorophyll a (a), chlorophyll b (b), total carotenoids (c), soluble sugar (d), proline(e), MDA (f), H2O2 (g), O2- (h) and activity of SOD (i), POD (j), SOD (k) of tobacco lines under heat stress; Each bar in the graph corresponds to the mean value ± SD of three independent replicates. * P < 0.05; ** P < 0.01.

Point 4:

In silico analysis of DfRAF1 can be improved.

Response 4:

The full length of dfraf1 gene is obtained by pre - transcriptional sequencing in the laboratory, so there is no need for in silico cloning analysis

Point 5:

Abstract:

The first line should be removed.

Response 5:

Thank you very much for your question and help us to improve the quality of our manuscript.

The first line has been removed.

Heat stress seriously affects medicinal herb growth and yield. Rubisco accumulation factor (Raf) is a key mediator regulating the activity of Rubisco, which play important roles in carbon assimilation and Calvin cycle in plants. Raf have been studied in many plants, but were rarely studied in the important medicinal Fragrant Woodfern (Dryopteris fragrans). The aim of this study was to analyze Raf on carbohydrate metabolism and response to heat stress in plants. In this study, high temperature treatment upregulated the expression of DfRaf, which was significantly higher than that of the DfPRK, DfRbcS, DfRbcL and DfRCA. The subcellular localization showed that the DfRaf protein were primarily located in the nucleus; DfPRK, DfRbcS, DfRbcL and DfRCA proteins were primarily located in the Chloroplast; we found that overexpressing DfRaf led to the increased activity of Rubisco, RCA and PRK under high temperature stress. Although the phenotype of DfRaf-OV-L2 and DfRaf-OV-L6 transgenic lines was not prominent, the H2O2, O2. and MDA content of DfRaf-OV-L2 and DfRaf-OV-L6 transgenic lines were significantly lower than those of WT and VC plants un-der high temperature stress. The photosynthetic pigments, proline, soluble sugar con-tent and ROS scavenging ability of DfRaf-OV-L2 and DfRaf-OV-L6 transgenic lines were higher than those of WT and VC plants under high temperature stress. The results showed that DfRaf could regulate carbon assimilation rate to enhance high temperature tolerance of plants by increasing Rubisco activity.

Point 6:

Introduction:

Point 6.1:

Line26: Both should be removed.

Response 6.1

Thank you very much for your question and help us to improve the quality of our manuscript.

Line26: Both has been removed.

Point 6.2:

Line37-40: sentence construction is wrong

Response 6.2

Thank you very much for your question and help us to improve the quality of our manuscript.

Sentence construction has been modified.

Rubisco is a key enzyme in the process of plant carbon assimilation. Its function is to catalyze the carboxylation of RuBP. However, Rubisco's catalytic efficiency is very low. It will interact with various phosphate sugars and lose its activity. For example, the substrate RuBP itself is a strong inhibitor of Rubisco. Therefore, the activated state of Rubisco is often an important factor determining the rate of carbon assimilation. In plants, the activation of Rubisco depends on the action of Rubisco activating enzyme (RCA). RCA releases the inhibitor bound to Rubisco by using the energy of ATP hydrolysis and changes the conformation of Rubisco, thereby stimulating its activity.

Point 6.3:

Line82-89: these lines are repeated twice.

Response 6.3:

Thank you very much for your question and help us to improve the quality of our manuscript.

Line82-89 have been modified.

Point 7:

Materials and Methods

Point 7.1:

How authors have collected the tissue from the control? Did the authors have collected at 0 hr and considered it as a control? or Did The authors collect at all the timepoint and pooled in for control tissue?

Response 7.1:

Thank you very much for your question and help us to improve the quality of our manuscript.

We selected the samples treated at high temperature for 0 hours (normal conditions) as the control.

Point 7.2:

What is the rationale of heat treatment? Why do plants directly shift to 42°C? provide the reference of the sudden temperature? Is 42°C too high for nicotiana?

Response 7.2:

Thank you very much for your question and help us to improve the quality of our manuscript.

For heat stress treatment, the potted plants were incubated at 42 °C under the light condition.

Fragrant Woodfern (Dryopteris fragrans) is an annual medicinal herb that grows in rock crevices and under wood [25] primarily in Wudalianchi City, Heilongjiang Province, China [26]. This region can experience significant temperature variability, with record high and low temperatures of 36.5°C and -41.0°C, respectively [27]. Therefore, we choose 42 degrees Celsius to treat plants.

It is reasonable to treat tobacco at a high temperature of 42 ℃. High temperature treatment refers to the method of Hamachi et al [29].

  1. Hamachi, A.; Nisihara, M.; Saito, S.; Rim, H.; Takahashi, H.; Islam, M.; Uemura, T.; Ohnishi, T.; Ozawa, R.; Maffei, M.E.; et al. Overexpression of geraniol synthase induces heat stress susceptibility in Nicotiana tabacum. Planta 2019, 249, 235-249, doi:10.1007/s00425-018-3054-z.

Point 7.3:

Line 105: for five times

Response 7.3:

Thank you very much for your question and help us to improve the quality of our manuscript.

“for five times” has been removed.

Tobacco seeds (Nicotiana tabacum) were surface sterilized for 3 min in 75% ethanol, rinsed with sterile water, and then germinated in 1/2 MS medium in a greenhouse with a 16 h-light and 8 h-dark photoperiod at 25 ± 2ËšC.

Point 7.4:

Line 107: plantletsfilled; These kind of merged words are throughout in the manuscript.

Response 7.4:

Following germination, the seedlings were transferred to a flowerpot containing 1:1 (V / V) highly nutritious soil and vermiculite and then cultured in a growth chamber with a 16 h-light and 8 h-dark photoperiod at 25 ± 2ËšC.

Point 7.5:

Line 96: Fragrant Woodfern (Dryopteris fragrans) seedlings were propagated from the laboratory of plant resources and molecular biology of Northeast Agricultural University. It should be procured instead of propagated.

Response 7.5:

Thank you very much for your question and help us to improve the quality of our manuscript.

“Propagated”has been replaced “procured”.

Fragrant Woodfern (Dryopteris fragrans) seedlings were procured from the laboratory of plant resources and molecular biology of Northeast Agricultural University.

Point 7.6:

Line 135-139: It shouldn’t be a part of materials and methods

Response 7.6:

Thank you very much for your question and help us to improve the quality of our manuscript.

Line 135-139 has been removed.

Point 8:

Results:

Point 8.1:

Rubisco large and small subunits have more than 1 copy in the genome. Did the authors try to look at the expression of other copies as well?

Response 8.1:

Thank you very much for your question and help us to improve the quality of our manuscript.

The genome of Fragrant Woodfern has not been detected yet. The laboratory pre transcriptome found that Rubisco large and small subunits genes are single transcripts. The copy numbers of Rubisco large and small subunits genes have not been found in other species (Tobacco, Arabidopsis thaliana, rice).

Point 8.2:

Section 3.2, one sentence has written in 6 lines with so many conjunctions and prepositions which is improper English and hard to follow for the reader.

Response 8.2:

Thank you very much for your question and help us to improve the quality of our manuscript.

In comparison with the 0h, the total Rubisco activity, initial Rubisco activity, and Rubisco activation state were significantly increased at 0.5 h (P < 0.05), but significantly decreased from 1-48 h (P < 0.01; Fig. 2a, Fig. 2b, and Fig. 2c); RCA and PRK activities were significantly increased from 0.5-1 h relative to 0 h (P < 0.05), but were significantly decreased from 3-48 h t (P < 0.01; Fig. 2d, and Fig. 2e).

Point 8.3:

Fig 2 b shows the rubisco activation state in low-temperature conditions. What is the relevance of this?

Response 8.3:

Thank you very much for your question and help us to improve the quality of our manuscript.

Fig 2 b has been modified

Fig. 2 Effects of high temperature stress on the carbon assimilation enzyme activity in the leaves of Fragrant Woodfern under different times. Effects of high temperature exposure on total Rubisco activity (a), initial Rubisco activity (b), Rubisco activation state (c), RCA activity (d) and PRK activity (e) in the leaves of Fragrant Woodfern under different times; Each bar in the graph corresponds to the mean value ± SD of three independent replicates. * P < 0.05; ** P < 0.01.

Point 8.4:

Fig 2 and 3, why authors did the Student’s t-test here? ANOVA will be appropriate for the data presented here.

Response 8.4:

Thank you very much for your question and help us to improve the quality of our manuscript.

Student’s t-test has been removed.

Point 8.5:

Fig 4 Authors have written DfPbcS and DfPbcL instead of DfRbcS and DfRbcL

Response 8.5:

Thank you very much for your question and help us to improve the quality of our manuscript.

Fig 4 has been modified

Fig. 4 Subcellular localization of DfRbcS, DfRbcL, DfRCA, DfRaf and DfPRK pro-teins. Subcellular localization of DfRbcS, DfRbcL, DfRCA, DfRaf and DfPRK protein in tobacco leaf cells, the red fluorescence of DAPI indicates the nucleus, bar =10μm.

Point 8.6:

Line278: “DfRaf was selected for the further study” instead of “DfRaf was selected for the next study”.

Response 8.6:

Thank you very much for your question and help us to improve the quality of our manuscript.

The manuscript has been modified

Point 8.7:

Line294: 35S:: DfRaf space should be omitted.

Response 8.7:

Thank you very much for your question and help us to improve the quality of our manuscript.

The manuscript has been modified

Point 8.8:

Section 3.5: There is no clearcut information on how authors have got Raf proteins from other species. How many RAF copies are in Fragrant Woodfern? The authors should conduct a genome-wide analysis for RAF.

Response 8.8:

Thank you very much for your question and help us to improve the quality of our manuscript.

2.11 Phylogenetic, multiple sequence alignment, and conservative motif analyses

Fragrant Woodfern Polypeptide sequences similar to Raf (MW524053) were identified using BLAST (http://blast.ncbi.nlm.nih.gov). The plant sequences used in the phylogenetic analysis were as follows: Physcomitrium patens (XP_024402609.1), Vitis vinifera (RVW88115.1), Carex littledalei (KAF3328238.1), Jatropha curcas (XP_012089314.1), Populus trichocarpa (XP_002319651.1), Vigna radiata (XP_014510931.1), Cicer arietinum (XP_004495565.1), Cajanus cajan (XP_020203342.1), Trema orientale (PON73478.1), Juglans regia (XP_018824157.2), Solanum pennellii (XP_015058205.1), Arachis hypogaea (XP_025655384.1), Eutrema salsugineum (XP_006408180.1), Selaginella moellendorffii (XP_024536106.1) and Arabidopsis thaliana (Q9SR19.1). Alignment was performed using CLUSTALW (http://genome.jp/tools/clustalw). A maximum-likelihood approach was used to construct an unrooted phylogenetic tree based upon amino acid sequences in MEGA 7 (Parameter settings: JTT model, Gamma distribution of site replacement rate, with 1000 Bootstrap reiterations). The MEME software (http://meme-suite.org/) was additionally used to predict conserved motifs, using default parameters with the exception of the maximum number of motifs being set to 10 (Parameter setting: optimal amino acid residue width 6-50; any number of repetitions; maximum 10 more motifs). DNAMAN was used for multiple sequence alignment of each DfRaf.

The whole gene of Dryopteris fragrans has not been sequenced.

Point 8.9:

The quality of the phylogram is very poor. Authors can consider the following paper for phylogram and genome-wide analysis. The figure needs to be reconstructed

Tiwari, M., Yadav, M., Singh, B., Pandey, V., Nawaz, K. and Bhatia, S. (2021). Evolutionary and functional analysis reveals CaRR13, a TypeB RR as positive regulator of symbiosis. Plant Biotechnol. J.

Tiwari, M., Pandey, V., Singh, B., Yadav, M. and Bhatia, S. (2021) Evolutionary and expression dynamics of LRR-RLKs and functional establishment of KLAVIER homolog in shoot mediated regulation of AON in chickpea symbiosis. Genomics

Pareek A., Mishra, D., Shekhar, S., Chakraborty, S., & Chakraborty, N. (2021). The small heat shock proteins, chaperonin 10, in plants: An evolutionary view and emerging functional diversity. Environmental and Experimental Botany, 182, 0432

Mishra, D., Shekhar, S., Chakraborty, S. and Chakraborty, N. (2018) Carboxylate clamp tetratricopeptide repeat (TPR) domain containing Hsp90 cochaperones in Triticeace: an insight into structural and functional diversification. Environ. Exp. Bot. 155, 31– 44

Mishra D., Suri G.S., Kaur G., Tiwari M. (2021): Comprehensive analysis of structural, functional, and evolutionary dynamics of Leucine Rich Repeats-RLKs in Thinopyrum elongatum. International Journal of Biological Macromolecules, 183: 513–527. https://doi.org/10.1016/j.ijbiomac.2021.04.137

Response 8.9:

Thank you very much for your question and help us to improve the quality of our manuscript.

The whole gene of Dryopteris fragrans has not been sequenced.

Fig. 5 has been reconstructed.

Fig. 5 Phylogenetic tree and conserved Motif analyses for the DfRaf protein. A maximum-likelihood approach was used to construct a phylogenetic tree based upon amino acid sequences using MEGA 7 (Parameter settings: JTT model, Gamma distributions of site replacement rate, with 1,000 Bootstrap replicate interations).

Point 8.10:

There is no reference for this “soluble sugar are an important osmotic regulator and nutrient. Its increase and accumulation protect biofilms” why the authors have chosen to do soluble sugar estimation in stress. Is there is any reference regarding that? Previously few studies have been performed to estimate soluble sugar in stress and developmental stages such as

Shekhar, S., Mishra, D., Buragohain, A. K., Chakraborty, S., & Chakraborty, N. (2015). Comparative analysis of phytochemicals and nutrient availability in two contrasting cultivars of sweet potato (Ipomoea batatas L.). Food Chemistry, 173, 957–965.

Mishra, D., Shekhar, S., Chakraborty, S., Chakraborty, N. (2021). Wheat 2-Cys peroxiredoxin plays a dual role in chlorophyll biosynthesis and adaptation to high temperatures. Plant J. 105,1374-1389

Response 8.10:

Thank you very much for your question and help us to improve the quality of our manuscript.

Soluble sugar are an important osmotic regulator and nutrient. Its increase and accumulation protect biofilms [36-39].

  1. Tian, Y.; Peng, K.; Bao, Y.; Zhang, D.; Meng, J.; Wang, D.; Wang, X.; Cang, J. Glucose-6-phosphate dehydrogenase and 6-phosphogluconate dehydrogenase genes of winter wheat enhance the cold tolerance of transgenic Arabidopsis. Plant Physiol Biochem 2021, 161, 86-97, doi:10.1016/j.plaphy.2021.02.005.
  2. Shekhar, S.; Mishra, D.; Buragohain, A.K.; Chakraborty, S.; Chakraborty, N. Comparative analysis of phytochemicals and nutrient availability in two contrasting cultivars of sweet potato (Ipomoea batatas L.). Food Chem 2015, 173, 957-965, doi:10.1016/j.foodchem.2014.09.172.
  3. Mishra, D.; Shekhar, S.; Chakraborty, S.; Chakraborty, N. Wheat 2-Cys peroxiredoxin plays a dual role in chlorophyll biosynthesis and adaptation to high temperature. Plant J 2021, 105, 1374-1389, doi:10.1111/tpj.15119.
  4. Ma, Y.Y.; Zhang, Y.L.; Lu, J.; Shao, H.B. Roles of plant soluble sugars and their responses to plant cold stress. African Journal of Biotechnology 2009, 8, 2004-2010.

Point 8.11:

There are no references for MDA and chlorophyll and no clear rationale why did authors perform all these assays. There are a lot of physiological studies conducted on many crops under heat stress. Authors can refer to them and make the connection in their manuscript.

Pareek A., Rathi D., Mishra, D., Shekhar, S., Chakraborty, S., & Chakraborty, N. (2019). Physiological plasticity to high temperature stress in chickpea: Adaptive responses and variable tolerance. Plant Sciences, 289, 110258.

Mishra, D., Shekhar S., Chakraborty S., Chakraborty N. (2017). Cultivar-specific high temperature stress responses in bread wheat (Triticum aestivum L.) associated with physicochemical traits and defense pathways. Food Chemistry, 221, 1077-1087

Response 8.11:

Thank you very much for your question and help us to improve the quality of our manuscript.

Malondialdehyde (MDA) is the main product of peroxidative degradation of mem-brane lipids, and it is also an important indicator to measure the degree of membrane lipid peroxidation [41,42].

  1. Mishra, D.; Shekhar, S.; Agrawal, L.; Chakraborty, S.; Chakraborty, N. Cultivar-specific high temperature stress responses in bread wheat (Triticum aestivum L.) associated with physicochemical traits and defense pathways. Food Chem 2017, 221, 1077-1087, doi:10.1016/j.foodchem.2016.11.053.
  2. Pareek, A.; Rathi, D.; Mishra, D.; Chakraborty, S.; Chakraborty, N. Physiological plasticity to high temperature stress in chickpea: Adaptive responses and variable tolerance. Plant Science 2019, 289, doi:10.1016/j.plantsci.2019.110258.

Point 8.12:

Fig 7 a. where is the control for genomic DNA?

Response 8.12:

Thank you very much for your question and help us to improve the quality of our manuscript.

-: ddH2O negative control, +: Positive control.

Point 8.13:

Fig 7 b. Why there is no marker in the picture? How much similarity between the DfRAF and NtRAF sequences? The authors have made primer for RT-PCR exclusively for DfRAF or is there some similarity between DfRAF and NtRAF sequences which is chosen for RT-PCR primers.

Response 8.13:

Thank you very much for your question and help us to improve the quality of our manuscript.

Fig. 7b is the verification of DfRaf expression in transgenic plants from the transcriptional level. The band size is known, so it is not necessary to label maker.

The sequence similarity between DfRaf and NtRaf was 40%. And the primer design of DfRaf is different from that of NtRaf (Fig. 1).

Fig. 1 Multiple sequence alignment for the DfRaf gene

Point 8.14:

Line 406: Herein; These are two separate words ‘Here’ and ‘in’. Please rectify throughout the manuscript.

Response 8.14:

Thank you very much for your question and help us to improve the quality of our manuscript.

Herein has been removed.

Point 8.15:

Line470: con-ditions

Response 8.15:

Thank you very much for your question and help us to improve the quality of our manuscript.

”con-ditions” has been modified.

Reviewer 2 Report

Comments for Authors

MS#genes-1731293

This work focuses on how overexpression of DfRaf gene regulates heat stress in tobacco plants. The function of Rubisco accumulation factor (Raf) is already known for its essential roles in mediating Rubisco assembly and carboxysome biogenesis in plants. In this study, the author showed significant unregulated expression of DfRaf gene that enhances heat stress “tolerance” in tobacco. After carefully checking this work, I found that this article has several major and minor limitations, such as:

Major concerns:

  1. Title, phenotypes of 2 DfRaf lines to heat stress, along with major findings and conclusion are not clear concerning this study.
  2. Title: does the author means “DfRaf lines enhances heat stress” or “enhances heat stress tolerance”?. The author must address and revise this issue.
  3. Plant phenotype of DfRaf lines under heat stress did not show any significant differences compared to WT and Vc lines ( 7C). The author must address the reason of it.
  4. Introduction part does not cover enough background information. Though the author wrote enough text. What was the need for current study with proper citation? The objectives and hypothesis are not clear. Authors should rework on the introduction to justify the work.
  5. Language correction is required throughout the manuscript by native speaker with track change mode.

Minor concerns

L1-3: title should be revised based on story/finding(s)

L7-18: Consequences should be maintained for “aim of the study, main results, interoperation and suggestion(s)”

L12, Please replace “. (dot) with and” by “, whereas”

L-16-18: Clear suggestion/findings should be provided associated with this study

L-Neither coherence nor research gap are properly addressed in the introduction section. So, the author should revise it properly.

-It is not wise to start a sentence with an abbreviation. Please carefully check several sections of this article, and write the full name at the start of the sentence.

-Table 1: gene ID does not specify gene name, please provide valid accession number/ID for all genes used in this study based on NCBI database.

L44, 348, 353: no need to mention significant level (p<0.01) with figure number. Please revise it.

-Conclusion is too short and lacks coherence between main findings and suggestion(s) and future recommendations.

Author Response

Response to Reviewer 1 Comments

MS#genes-1731293

This work focuses on how overexpression of DfRaf gene regulates heat stress in tobacco plants. The function of Rubisco accumulation factor (Raf) is already known for its essential roles in mediating Rubisco assembly and carboxysome biogenesis in plants. In this study, the author showed significant unregulated expression of DfRaf gene that enhances heat stress “tolerance” in tobacco. After carefully checking this work, I found that this article has several major and minor limitations, such as:

Major concerns:

Point 1:

  1. Title, phenotypes of 2 DfRaf lines to heat stress, along with major findings and conclusion are not clear concerning this study.

Response 1:

Thank you very much for your question and help us to improve the quality of our manuscript.

we found that high temperature treatment upregulated the expression of DfRaf. Overexpressing DfRaf led to the in-creased activity of Rubisco, RCA and PRK under high temperature stress. The photosynthetic pigments, proline, soluble sugar content and ROS scavenging ability of DfRaf-OV-L2 and DfRaf-OV-L6 transgenic lines were higher than those of WT and VC plants under high temperature stress. The results showed that DfRaf could regulate carbon assimilation rate to enhance high tempera-ture tolerance of plants by increasing Rubisco activity.

Point 2:

  1. Title: does the author means “DfRaf lines enhances heat stress” or “enhances heat stress tolerance”?. The author must address and revise this issue.

Response 2:

Thank you very much for your question and help us to improve the quality of our manuscript.

Title has been modified.

Overexpression of DfRaf from Fragrant Woodfern (Dryopteris fragrans) Enhances high temperature tolerance in tobacco (Nicotiana tabacum)

Point 3:

  1. Plant phenotype of DfRaf lines under heat stress did not show any significant differences compared to WT and Vc lines ( 7C). The author must address the reason of it.

Response 3:

Thank you very much for your question and help us to improve the quality of our manuscript.

we found that overexpressing DfRaf led to the increased activity of Rubisco, RCA and PRK under high temperature stress. Although the phenotype of DfRaf-OV-L2 and DfRaf-OV-L6 transgenic lines was not prominent, the H2O2, O2. and MDA content of DfRaf-OV-L2 and DfRaf-OV-L6 transgenic lines were significantly lower than those of WT and VC plants un-der high temperature stress. The photosynthetic pigments, proline, soluble sugar con-tent and ROS scavenging ability of DfRaf-OV-L2 and DfRaf-OV-L6 transgenic lines were higher than those of WT and VC plants under high temperature stress. The results showed that DfRaf could regulate carbon assimilation rate to enhance high temperature tolerance of plants by increasing Rubisco activity.

Point 4:

  1. Introduction part does not cover enough background information. Though the author wrote enough text. What was the need for current study with proper citation? The objectives and hypothesis are not clear. Authors should rework on the introduction to justify the work.

Response 4:

Thank you very much for your question and help us to improve the quality of our manuscript.

Introduction has been modified.

Plants can grow and develop normally within a certain temperature range. When the temperature is higher or lower than the critical temperature required for growth, however, plant growth can be delayed or inhibited [1]. Temperature is one of the most important natural determinants of plant photosynthetic activity, and over 90% of plant dry weight is derived from photosynthesis [2]. High temperatures can adversely impact photosynthetic rates, energy metabolism, chloroplast submicroscopic structural integrity, photosynthetic pigment content, and other key biochemical processes necessary to plant survival [3,4]. High temperatures not only cause changes in photosynthetic pigments and transpiration, but also substantially alter the activities of Calvin cycle-related enzymes [5,6], including ribulose-1,5-bisphosphate carboxylase/oxygenase (Rubisco), which is the key carboxylase involved in plant C3 photosynthetic processes [7]. Rubisco is a hexadecameric complex composed of eight large subunits (RbcL, 50 - 55 kDa) and eight small subunits (RbcS, 12 - 18 kDa), denoted as RbcL8S8 [8]. RbcL subunits are arranged as a tetramer of antiparallel RbcL dimers such that RbcL8 serves as the primary catalytic core, with eight RbcS subunits that dock at the top and bottom to stabilize RbcL8 and tune Rubisco activity [9]. Rubisco is a key enzyme in the process of plant carbon assimilation. Its function is to catalyze the carboxylation of RuBP. However, Rubisco's catalytic efficiency is very low. It will interact with various phosphate sugars and lose its activity. For example, the substrate RuBP itself is a strong inhibitor of Rubisco [10]. Therefore, the activated state of Rubisco is often an important factor determining the rate of carbon assimilation. In plants, the activation of Rubisco depends on the action of Rubisco activating enzyme (RCA). RCA releases the inhibitor bound to Rubisco by using the energy of ATP hydrolysis and changes the conformation of Rubisco, thereby stimulating its activity [11]. In wheat (Triticum aestivum), significant increases in RCA activity enable Rubisco to maintain high CO2 assimilation efficiency [12]. As a molecular chaperone for Rubisco [13], RCA can control and stabilize Rubisco activity and thereby regulate photosynthesis under high temperature stress [14]. In Zea mays, another RbcL-interacting chaperone termed Rubisco accumulation factor (Raf) was identified [8,15]. Raf1 has been shown to promote recombinant Rubisco assembly in tobacco (Nicotiana tabacum) chloroplasts, and its overexpression can bolster Rubisco protein expression and activation [16]. Phosphoribokinase (PRK) is a critical regulator of carbon assimilation and controls the homeostatic balance between assimilation and regeneration activities [17]. PRK can catalyze Ribulose-1,5-bisphosphate (RuBP) regeneration while providing a receptor for CO2 assimilation [18]. In studies wherein Rubisco levels in tobacco plants were decreased, reduced photosynthesis and growth were observed [19,20], with similar results also being found in rice plants [21]. Conversely, increases in the activity levels of Rubisco resulted in increased photosynthesis and biomass in tobacco plants both under controlled conditions [22] as well as in the field under elevated CO2 [23]. In food crops such as tomatoes, increased RCA activities have been reported to result in similar increases in photosynthesis and growth, as well as improved chilling tolerance [24]. It indicates that the increase of carbon assimilation related enzyme activity promotes the enhancement of plant photosynthesis and then promotes plant growth and yield, especially in important food crop varieties. At the same time, these results provide evidence that increasing the activity of carbon assimilation related enzymes enhance high temperature tolerance in plant. However, the relationship between carbohydrate metabolism and response to high temperature stress needs to be further studied.

Fragrant Woodfern (Dryopteris fragrans) is an annual medicinal herb that grows in rock crevices and under wood [25] primarily in Wudalianchi City, Heilongjiang Province, China [26]. This region can experience significant temperature variability, with record high and low temperatures of 36.5°C and -41.0°C, respectively [27]. Fragrant Woodfern plants contain benzene triphenols, terpenes, flavonoids and phenylpropanoid compounds that exhibit inhibitory effects on fungi. These plants have also been used to alleviate skin itching, allergies and rheumatoid arthritis [28]. However, habitat reductions and excess utilization have led wild Fragrant Woodfern to become increasingly endangered. Further studies of the molecular mechanisms governing Fragrant Woodfern stress resistance will therefore be beneficial for the artificial cultivation of this plant necessary to meet consumer demand. Carbon assimilation is a key facet of the heat stress response pathways in plants. Therefore, Fragrant Woodfern is a valuable material for studying the heat stress tolerance function of carbon assimilation related genes in ferns. We explored changes in carbon assimilation enzyme activity (including Rubisco activation state, total Rubisco activity, initial Rubisco activity, RCA and PRK activity), and relative levels of genes associated with carbon assimilation including DfPRK, DfRbcS, DfRbcL, DfRCA and DfRaf were assessed via quantitative real-time PCR (qRT-PCR). we isolated and identified a DfRaf gene from Fragrant Woodferns. Ectopic expression of DfRaf enhanced heat stress tolerance in tobacco (Nicotiana tabacum). These results demonstrate that DfRaf acts as a positive role in heat resistance, which may provide a reference for breeding heat resistant cultivars in future genetic engineering experiment.

Point 5:

  1. Language correction is required throughout the manuscript by native speaker with track change mode.

Response 5:

Thank you very much for your question and help us to improve the quality of our manuscript.

The article has been polished by the polishing company.

Minor concerns

Point 6:

L1-3: title should be revised based on story/finding(s)

Response 6:

Thank you very much for your question and help us to improve the quality of our manuscript.

Title has been modified.

Overexpression of DfRaf from Fragrant Woodfern (Dryopteris fragrans) Enhances high temperature tolerance in tobacco (Nicotiana tabacum)

Point 7:

L7-18: Consequences should be maintained for “aim of the study, main results, interoperation and suggestion(s)”

Response 7:

Thank you very much for your question and help us to improve the quality of our manuscript.

Abstract has been modified

Heat stress seriously affects medicinal herb growth and yield. Rubisco accumulation factor (Raf) is a key mediator regulating the activity of Rubisco, which play important roles in carbon assimilation and Calvin cycle in plants. Raf have been studied in many plants, but were rarely studied in the important medicinal Fragrant Woodfern (Dryopteris fragrans). The aim of this study was to analyze Raf on carbohydrate metabolism and response to heat stress in plants. In this study, high temperature treatment upregulated the expression of DfRaf, which was significantly higher than that of the DfPRK, DfRbcS, DfRbcL and DfRCA. The subcellular localization showed that the DfRaf protein were primarily located in the nucleus; DfPRK, DfRbcS, DfRbcL and DfRCA proteins were primarily located in the Chloroplast; we found that overexpressing DfRaf led to the increased activity of Rubisco, RCA and PRK under high temperature stress. Although the phenotype of DfRaf-OV-L2 and DfRaf-OV-L6 transgenic lines was not prominent, the H2O2, O2. and MDA content of DfRaf-OV-L2 and DfRaf-OV-L6 transgenic lines were significantly lower than those of WT and VC plants un-der high temperature stress. The photosynthetic pigments, proline, soluble sugar con-tent and ROS scavenging ability of DfRaf-OV-L2 and DfRaf-OV-L6 transgenic lines were higher than those of WT and VC plants under high temperature stress. The results showed that DfRaf could regulate carbon assimilation rate to enhance high temperature tolerance of plants by increasing Rubisco activity.

Point 8:

L12, Please replace “. (dot) with and” by “, whereas”

Response 8:

Thank you very much for your question and help us to improve the quality of our manuscript.

L12 has been modified.

Point 9:

L-16-18: Clear suggestion/findings should be provided associated with this study

Response 9:

Thank you very much for your question and help us to improve the quality of our manuscript.

L12 has been modified

Point 10:

L-Neither coherence nor research gap are properly addressed in the introduction section. So, the author should revise it properly.

Response 10:

Thank you very much for your question and help us to improve the quality of our manuscript.

Introduction has been modified.

Plants can grow and develop normally within a certain temperature range. When the temperature is higher or lower than the critical temperature required for growth, however, plant growth can be delayed or inhibited [1]. Temperature is one of the most important natural determinants of plant photosynthetic activity, and over 90% of plant dry weight is derived from photosynthesis [2]. High temperatures can adversely impact photosynthetic rates, energy metabolism, chloroplast submicroscopic structural integrity, photosynthetic pigment content, and other key biochemical processes necessary to plant survival [3,4]. High temperatures not only cause changes in photosynthetic pigments and transpiration, but also substantially alter the activities of Calvin cycle-related enzymes [5,6], including ribulose-1,5-bisphosphate carboxylase/oxygenase (Rubisco), which is the key carboxylase involved in plant C3 photosynthetic processes [7]. Rubisco is a hexadecameric complex composed of eight large subunits (RbcL, 50 - 55 kDa) and eight small subunits (RbcS, 12 - 18 kDa), denoted as RbcL8S8 [8]. RbcL subunits are arranged as a tetramer of antiparallel RbcL dimers such that RbcL8 serves as the primary catalytic core, with eight RbcS subunits that dock at the top and bottom to stabilize RbcL8 and tune Rubisco activity [9]. Rubisco is a key enzyme in the process of plant carbon assimilation. Its function is to catalyze the carboxylation of RuBP. However, Rubisco's catalytic efficiency is very low. It will interact with various phosphate sugars and lose its activity. For example, the substrate RuBP itself is a strong inhibitor of Rubisco [10]. Therefore, the activated state of Rubisco is often an important factor determining the rate of carbon assimilation. In plants, the activation of Rubisco depends on the action of Rubisco activating enzyme (RCA). RCA releases the inhibitor bound to Rubisco by using the energy of ATP hydrolysis and changes the conformation of Rubisco, thereby stimulating its activity [11]. In wheat (Triticum aestivum), significant increases in RCA activity enable Rubisco to maintain high CO2 assimilation efficiency [12]. As a molecular chaperone for Rubisco [13], RCA can control and stabilize Rubisco activity and thereby regulate photosynthesis under high temperature stress [14]. In Zea mays, another RbcL-interacting chaperone termed Rubisco accumulation factor (Raf) was identified [8,15]. Raf1 has been shown to promote recombinant Rubisco assembly in tobacco (Nicotiana tabacum) chloroplasts, and its overexpression can bolster Rubisco protein expression and activation [16]. Phosphoribokinase (PRK) is a critical regulator of carbon assimilation and controls the homeostatic balance between assimilation and regeneration activities [17]. PRK can catalyze Ribulose-1,5-bisphosphate (RuBP) regeneration while providing a receptor for CO2 assimilation [18]. In studies wherein Rubisco levels in tobacco plants were decreased, reduced photosynthesis and growth were observed [19,20], with similar results also being found in rice plants [21]. Conversely, increases in the activity levels of Rubisco resulted in increased photosynthesis and biomass in tobacco plants both under controlled conditions [22] as well as in the field under elevated CO2 [23]. In food crops such as tomatoes, increased RCA activities have been reported to result in similar increases in photosynthesis and growth, as well as improved chilling tolerance [24]. It indicates that the increase of carbon assimilation related enzyme activity promotes the enhancement of plant photosynthesis and then promotes plant growth and yield, especially in important food crop varieties. At the same time, these results provide evidence that increasing the activity of carbon assimilation related enzymes enhance high temperature tolerance in plant. However, the relationship between carbohydrate metabolism and response to high temperature stress needs to be further studied.

Fragrant Woodfern (Dryopteris fragrans) is an annual medicinal herb that grows in rock crevices and under wood [25] primarily in Wudalianchi City, Heilongjiang Province, China [26]. This region can experience significant temperature variability, with record high and low temperatures of 36.5°C and -41.0°C, respectively [27]. Fragrant Woodfern plants contain benzene triphenols, terpenes, flavonoids and phenylpropanoid compounds that exhibit inhibitory effects on fungi. These plants have also been used to alleviate skin itching, allergies and rheumatoid arthritis [28]. However, habitat reductions and excess utilization have led wild Fragrant Woodfern to become increasingly endangered. Further studies of the molecular mechanisms governing Fragrant Woodfern stress resistance will therefore be beneficial for the artificial cultivation of this plant necessary to meet consumer demand. Carbon assimilation is a key facet of the heat stress response pathways in plants. Therefore, Fragrant Woodfern is a valuable material for studying the heat stress tolerance function of carbon assimilation related genes in ferns. We explored changes in carbon assimilation enzyme activity (including Rubisco activation state, total Rubisco activity, initial Rubisco activity, RCA and PRK activity), and relative levels of genes associated with carbon assimilation including DfPRK, DfRbcS, DfRbcL, DfRCA and DfRaf were assessed via quantitative real-time PCR (qRT-PCR). we isolated and identified a DfRaf gene from Fragrant Woodferns. Ectopic expression of DfRaf enhanced heat stress tolerance in tobacco (Nicotiana tabacum). These results demonstrate that DfRaf acts as a positive role in heat resistance, which may provide a reference for breeding heat resistant cultivars in future genetic engineering experiment.

Point 11:

-It is not wise to start a sentence with an abbreviation. Please carefully check several sections of this article, and write the full name at the start of the sentence.

Response 11:

Thank you very much for your question and help us to improve the quality of our manuscript.

Manuscript has been modified.

Point 12:

-Table 1: gene ID does not specify gene name, please provide valid accession number/ID for all genes used in this study based on NCBI database.

Response 12:

Thank you very much for your question and help us to improve the quality of our manuscript.

The GenBank accession numbers for DfPRK, DfRbcS, DfRbcL, DfRCA and DfRaf were MW524052, MW524050, MW524049, MW524051 and MW524053, respectively. Primers used in this study are compiled in Table 1.

Table 1 Primers for carbon assimilation related genes

Gene ID

GenBank

Forward primer (5' to 3')

Reverse primer (5' to 3')

Purpose

DfRbcL

MW524049

GGAGACTAAAGCAGGTGTTG

GTCAACCCATCTGTCCATAC

qRT-PCR

DfRbcS

MW524050

ACTGGTAAGGACAGGTGGAA

CCATGAAAGCAGGAGCAA

qRT-PCR

DfRCA

MW524051

CGTCCACTGCCACTACTCCT

TCGTAAGCAAGGCCACCC

qRT-PCR

DfPRK

MW524052

CTGCCCTCCCTTCTTCGT

GCAGCACCTCCGAAGACA

qRT-PCR

DfRaf

MW524053

TGGGCTGTGGTTTGACTA

AGGCGAGCATACTTTCTTC

qRT-PCR

Df18sRNA

GCTTTCGCAGTAGTTCGTCTTTC

TGGTCCTATTATGTTGGTCTTCGG

qRT-PCR

DfRbcL

ATGTCACCACAAACGGAGACT

TTACAACGTATCAATTGTCTCG

Gene clone

DfRbcS

ATGGCTACTACTGTAGCTGCT

CTCCCGCACTCATCATGATTG

Gene clone

DfRCA

ATGGCGTCCACTGCCACTACTC

TCACTTGGGGTCTGCAAAATCA

Gene clone

DfPRK

ATGAAACGCATTCACATATTC

TCAAGCCTTAGTGGATTGCAAA

Gene clone

DfRaf

ATGGCGTTTTTGGCGGTCTCA

TTAGTCCCATTCTATAGCATTC

Gene clone

DfRbcL-EGFP

CTCGGTACCCGGGGATCC

ATGTCACCACAAACGGAGACT

GGTGTCGACTCTAGAGGATCC

TTACAACGTATCAATTGTCTCG

Gene clone

DfRbcS-EGFP

CTCGGTACCCGGGGATCC ATGGCTACTACTGTAGCTGCT

GGTGTCGACTCTAGAGGATCC CTCCCGCACTCATCATGATTG

Gene clone

DfRCA-EGFP

CTCGGTACCCGGGGATCC ATGGCGTCCACTGCCACTACTC

GGTGTCGACTCTAGAGGATCC TCACTTGGGGTCTGCAAAATCA

Gene clone

DfPRK-EGFP

CTCGGTACCCGGGGATCC ATGAAACGCATTCACATATTC

GGTGTCGACTCTAGAGGATCC TCAAGCCTTAGTGGATTGCAAA

Gene clone

DfRaf-EGFP

CTCGGTACCCGGGGATCC ATGGCGTTTTTGGCGGTCTCA

GGTGTCGACTCTAGAGGATCC TTAGTCCCATTCTATAGCATTC

Gene clone

pCAMBIA2301- DfRaf

GGGCATCGATACGGGATCCAT

ATGGCGTTTTTGGCGGTCTCA

TCGAGCTCGATGGATCCCGTA

TTAGTCCCATTCTATAGCATTC

Gene clone

Point 13:

L44, 348, 353: no need to mention significant level (p<0.01) with figure number. Please revise it.

Response 13:

Thank you very much for your question and help us to improve the quality of our manuscript.

L44, 348, 353 have been modified.

Point 14:

-Conclusion is too short and lacks coherence between main findings and suggestion(s) and future recommendations.

Response 14:

Thank you very much for your question and help us to improve the quality of our manuscript.

Conclusion has been modified.

we found that high temperature treatment upregulated the expression of DfRaf. Overexpressing DfRaf led to the increased activity of Rubisco, RCA and PRK under high temperature stress. The photosynthetic pigments, proline, soluble sugar content and ROS scavenging ability of DfRaf-OV-L2 and DfRaf-OV-L6 transgenic lines were higher than those of WT and VC plants under high temperature stress. The results showed that DfRaf could regulate carbon assimilation rate to enhance high temperature tolerance of plants by increasing Rubisco activity.

Round 2

Reviewer 1 Report

Line 13: change to medicinal plant.

A lot of acronyms were used in abstract which could make it hard to be followed by general audience.

Change Line 41 to 43 as: Temperature higher (heat stress) or lower (cold stress) than the optimum temperature required for plant growth can inhibit or delay the growth. Consider citing recent articles (recommended but not mandatory)
https://onlinelibrary.wiley.com/doi/full/10.1002/cche.10523, https://www.sciencedirect.com/science/article/pii/B9780128193341000095 and https://onlinelibrary.wiley.com/doi/full/10.1111/pce.14266

Line 124: change we to We.

Line 126: replace acts as with plays.

Line 134: remove and

Line 158, 159: 5U3-phosphorus Acid glyceraldehyde dehydrogenase, 5U3-phosphoglycerate phosphokinase, 17.5U phospho- creatine kinase. What are these?

Line 191, 192: 2 Uml pyruvate kinase, 2 Uml lactate dehydrogen- 191 ase, 1.5 Uml 1 ribose-5-phosphate (R5P)-isomerase. What is Uml?

In Table 1 replace gene clone with gene cloning

Line 240: H2O2- change to H2O2.

Line 245: Change the sentence it should not start with Put.

Line 247: Remove then.

â—¦C is not uniform throughout the manuscript for ex. check Line 247 and 248.

Line 260: Polypeptide to polypeptide.

Line 261: Mention the e-value of Blast.

Line 261: change sentence to The Raf protein sequences used from other plant species for phylogenetic analysis were as follows (recommended but not mandatory):

Better this section as mentioned in the methodology section of articles, https://onlinelibrary.wiley.com/doi/full/10.1111/pbi.13649, https://www.sciencedirect.com/science/article/pii/S0888754321004043, https://www.sciencedirect.com/science/article/pii/S014181302100903X   Consider citing these references.

Line 305: Change 0h to 0 h and check it throughout manuscript.

Line 309: 3-48 h t. What is t here?

Line 394: Remove  were

Line 395: Does author mean DfRAf instead of DfRCA

Line 397: semi to Semi

Line 398: Change to “that levels of DfRaf transcripts in DfRaf-OV-L2 and DfRaf-OV-L6 was significantly higher.

Line 403, 404: Repetition, remove one of the sentence.

Line 408: remove it in

Line 427: ROS will accumulate rapidly in plant, and high concentrations of ROS will directly affect the normal development of cells. Reference missing consider citing (recommended but not mandatory) https://www.ncbi.nlm.nih.gov/pmc/articles/PMC8784697/

Line 431, 433: Remove it in

Line 559: in-creased to increased

Author Response

Response to Reviewer 1 Comments

Point 1:

Line 13: change to medicinal plant.

Response 1:

Thank you very much for your question and help us to improve the quality of our manuscript.

Abstract have been modified.

The aim of this study was to analyze Raf on carbohydrate metabolism and response to heat stress in medicinal plants.

Point 2:

Line 13

A lot of acronyms were used in abstract which could make it hard to be followed by general audience.

Response 2:

Thank you very much for your question and help us to improve the quality of our manuscript.

Abstract have been modified.

Abstract: Heat stress seriously affects medicinal herb growth and yield. Rubisco accumulation factor (Raf) is a key mediator regulating the activity of Ribulose-1,5-bisphosphate carboxylase/oxygenase (Rubisco), which play important roles in carbon assimilation and Calvin cycle in plants. Raf have been studied in many plants, but were rarely studied in the important medicinal Fragrant Woodfern (Dryopteris fragrans). The aim of this study was to analyze Raf on carbohydrate metabolism and response to heat stress in medicinal plants. In this study, high temperature treatment upregulated the expression of DfRaf, which was significantly higher than that of the phosphoribokinase (DfPRK), Rubisco small subunits (DfRbcS), Rubisco large subunits (DfRbcL) and Rubisco activase (DfRCA). The subcellular localization showed that the DfRaf protein were primarily located in the nucleus; DfPRK, DfRbcS, DfRbcL and DfRCA proteins were primarily located in the Chloroplast; we found that overexpressing DfRaf led to the increased activity of Rubisco, RCA and PRK under high temperature stress. The H2O2, O2. and MDA content of DfRaf-OV-L2 and DfRaf-OV-L6 transgenic lines were significantly lower than those of WT and VC plants under high temperature stress. The photosynthetic pigments, proline, soluble sugar content and ROS scavenging ability of DfRaf-OV-L2 and DfRaf-OV-L6 transgenic lines were higher than those of WT and VC plants under high temperature stress. The results showed that overexpression of DfRaf gene increased the Rubisco activity, which enhance high temperature tolerance of plants.

Point 3:

Change Line 41 to 43 as: Temperature higher (heat stress) or lower (cold stress) than the optimum temperature required for plant growth can inhibit or delay the growth. Consider citing recent articles (recommended but not mandatory)

https://onlinelibrary.wiley.com/doi/full/10.1002/cche.10523, https://www.sciencedirect.com/science/article/pii/B9780128193341000095 and https://onlinelibrary.wiley.com/doi/full/10.1111/pce.14266

Response 3:

Thank you very much for your question and help us to improve the quality of our manuscript.

Manuscript have been modified.

Temperature higher (heat stress) or lower (cold stress) than the optimum temperature required for plant growth can inhibit or delay the growth [1-4].

  1. Sukhov, V.; Gaspirovich, V.; Mysyagin, S.; Vodeneev, V. High-Temperature Tolerance of Photosynthesis Can Be Linked to Local Electrical Responses in Leaves of Pea. Front Physiol 2017, 8, 763, doi:10.3389/fphys.2017.00763.
  2. Hein, N.T.; Impa, S.M.; Wagner, D.; Bheemanahalli, R.; Kumar, R.; Tiwari, M.; Prasad, P.V.V.; Tilley, M.; Wu, X.; Neilsen, M.; et al. Grain micronutrient composition and yield components in field-grown wheat are negatively impacted by high night-time temperature. Cereal Chemistry 2022, 99, 615-624, doi:10.1002/cche.10523.
  3. Tiwari, M.; Kumar, R.; Min, D.; Jagadish, S.V.K. Genetic and molecular mechanisms underlying root architecture and function under heat stress-A hidden story. Plant Cell and Environment 2022, 45, 771-788, doi:10.1111/pce.14266.
  4. A, P.S.; B, M.M.M.S.; C, A.P.; D, M.V.; C, S.M.; E, B.S.; E, M.T.; E, V.P. The role of key transcription factors for cold tolerance in plants. Transcription Factors for Abiotic Stress Tolerance in Plants 2020, 123-152, doi:https://doi.org/10.1016/B978-0-12-819334-1.00009-5.

Point 4:

Line 124: change we to We.

Response 4:

Thank you very much for your question and help us to improve the quality of our manuscript.

Manuscript have been modified.

We isolated and identified a DfRaf gene from Fragrant Woodferns. Ectopic expression of DfRaf enhanced heat stress tolerance in tobacco (Nicotiana tabacum).

Point 5:

Line 126: replace acts as with plays.

Response 5:

Thank you very much for your question and help us to improve the quality of our manuscript.

Manuscript have been modified.

These results demonstrate that DfRaf plays a positive role in heat resistance, which may provide a reference for breeding heat resistant cultivars in future genetic engineering experiment.

Point 6:

Line 134: remove and

Response 6:

Thank you very much for your question and help us to improve the quality of our manuscript.

Manuscript have been modified.

180 pots of 12-week-old seeds (60 in each of the 3 replicates) with consistent growth then were transferred to a 42°C growth room to simulate high temperature stress conditions, respectively

Point 7:

Line 158, 159: 5U3-phosphorus Acid glyceraldehyde dehydrogenase, 5U3-phosphoglycerate phosphokinase, 17.5U phospho- creatine kinase. What are these?

Response 7:

Thank you very much for your question and help us to improve the quality of our manuscript.

Manuscript have been modified.

Change “5U3-phosphorus Acid glyceraldehyde dehydrogenase” to “5 U 3-phosphorus acid glyceraldehyde dehydrogenase”

Change “5U3-phosphoglycerate phosphokinase” to “5 U 3-phosphoglycerate phosphokinase”

Change “17.5U phosphocreatine kinase” to “17.5 U phosphocreatine kinase”

Point 8:

Line 191, 192: 2 Uml pyruvate kinase, 2 Uml lactate dehydrogen- 191 ase, 1.5 Uml 1 ribose-5-phosphate (R5P)-isomerase. What is Uml?

Response 8:

Thank you very much for your question and help us to improve the quality of our manuscript.

Change “2 Uml pyruvate kinase, 2 Uml lactate dehydrogen- 191 ase, 1.5 Uml 1 ribose-5-phosphate (R5P)-isomerase” to “2 U.mL-1 pyruvate kinase, 2 U.mL-1 lactate dehydrogenase, 1.5 U.mL-1 1 ribose-5-phosphate (R5P)-isomerase”

U.mL-1 is the unit of enzyme activity

Point 9:

In Table 1 replace gene clone with gene cloning

Response 9:

Thank you very much for your question and help us to improve the quality of our manuscript.

Table 1 have been modified.

Table 1 Primers for carbon assimilation related genes

Gene ID

GenBank

Forward primer (5' to 3')

Reverse primer (5' to 3')

Purpose

DfRbcL

MW524049

GGAGACTAAAGCAGGTGTTG

GTCAACCCATCTGTCCATAC

qRT-PCR

DfRbcS

MW524050

ACTGGTAAGGACAGGTGGAA

CCATGAAAGCAGGAGCAA

qRT-PCR

DfRCA

MW524051

CGTCCACTGCCACTACTCCT

TCGTAAGCAAGGCCACCC

qRT-PCR

DfPRK

MW524052

CTGCCCTCCCTTCTTCGT

GCAGCACCTCCGAAGACA

qRT-PCR

DfRaf

MW524053

TGGGCTGTGGTTTGACTA

AGGCGAGCATACTTTCTTC

qRT-PCR

Df18sRNA

GCTTTCGCAGTAGTTCGTCTTTC

TGGTCCTATTATGTTGGTCTTCGG

qRT-PCR

DfRbcL

ATGTCACCACAAACGGAGACT

TTACAACGTATCAATTGTCTCG

gene cloning

DfRbcS

ATGGCTACTACTGTAGCTGCT

CTCCCGCACTCATCATGATTG

gene cloning

DfRCA

ATGGCGTCCACTGCCACTACTC

TCACTTGGGGTCTGCAAAATCA

gene cloning

DfPRK

ATGAAACGCATTCACATATTC

TCAAGCCTTAGTGGATTGCAAA

gene cloning

DfRaf

ATGGCGTTTTTGGCGGTCTCA

TTAGTCCCATTCTATAGCATTC

gene cloning

DfRbcL-EGFP

CTCGGTACCCGGGGATCC

ATGTCACCACAAACGGAGACT

GGTGTCGACTCTAGAGGATCC

TTACAACGTATCAATTGTCTCG

gene cloning

DfRbcS-EGFP

CTCGGTACCCGGGGATCC ATGGCTACTACTGTAGCTGCT

GGTGTCGACTCTAGAGGATCC CTCCCGCACTCATCATGATTG

gene cloning

DfRCA-EGFP

CTCGGTACCCGGGGATCC ATGGCGTCCACTGCCACTACTC

GGTGTCGACTCTAGAGGATCC TCACTTGGGGTCTGCAAAATCA

gene cloning

DfPRK-EGFP

CTCGGTACCCGGGGATCC ATGAAACGCATTCACATATTC

GGTGTCGACTCTAGAGGATCC TCAAGCCTTAGTGGATTGCAAA

gene cloning

DfRaf-EGFP

CTCGGTACCCGGGGATCC ATGGCGTTTTTGGCGGTCTCA

GGTGTCGACTCTAGAGGATCC TTAGTCCCATTCTATAGCATTC

gene cloning

pCAMBIA2301- DfRaf

GGGCATCGATACGGGATCCAT

ATGGCGTTTTTGGCGGTCTCA

TCGAGCTCGATGGATCCCGTA

TTAGTCCCATTCTATAGCATTC

gene cloning

Point 10:

Line 240: H2O2- change to H2O2.

Response 10:

Thank you very much for your question and help us to improve the quality of our manuscript.

The contents of proline, H2O2, O2., malondialdehyde, soluble sugar and the activities of superoxide dismutase, peroxidase and catalase in tobacco were determined by the proline content Kit (PRO-1-Y), H2O2 content Kit (H2O2-1-Y), O2. content Kit (SA-1-G), Malonic dialdehyde (MDA-1-Y), Plant soluble sugar content kit (KT-1-Y), Superoxide Dismutase Kit (SOD-1-Y), Peroxidase Kit (POD-1-Y), and Catalase Kit (CAT-1-W) according to the manufacturer’s instructions (Comin Biotechnology, Suzhou, China).

Point 11:

Line 245: Change the sentence it should not start with Put.

Response 11:

Thank you very much for your question and help us to improve the quality of our manuscript.

Fresh leaves are put into 1 mg.mL1 3,3′-diaminobenzidine (DAB) solution (pH 5.5, 50 mM Tris HCl) and 0.2% nitro blue tetrazolium (NBT) solution (pH 7.8, 50 mM phosphate buffer) respectively.

Point 12:

Line 247: Remove then.

Response 12:

Thank you very much for your question and help us to improve the quality of our manuscript.

Manuscript have been modified.

Point 13:

â—¦C is not uniform throughout the manuscript for ex. check Line 247 and 248.

Response 13:

Thank you very much for your question and help us to improve the quality of our manuscript.

The leaf samples were then placed in the dark at 28°C for 12 h. Finally, decolorization was carried out in a 70°C boiling water bath with 90% ethanol and stored in 50% glycerol.

Point 14:

Line 260: Polypeptide to polypeptide.

Response 14:

Thank you very much for your question and help us to improve the quality of our manuscript.

Change Polypeptide to polypeptide

Fragrant Woodfern polypeptide sequences similar to Raf (MW524053) were identified using BLAST (http://blast.ncbi.nlm.nih.gov).

Point 15:

Line 261: Mention the e-value of Blast.

Response 15:

Thank you very much for your question and help us to improve the quality of our manuscript.

Fragrant Woodfern polypeptide sequences similar to Raf (MW524053) were identified using BLAST (E < 1 × 105) (http://blast.ncbi.nlm.nih.gov) [38].

Point 16:

Line 261: change sentence to The Raf protein sequences used from other plant species for phylogenetic analysis were as follows (recommended but not mandatory):

Better this section as mentioned in the methodology section of articles, https://onlinelibrary.wiley.com/doi/full/10.1111/pbi.13649, https://www.sciencedirect.com/science/article/pii/S0888754321004043, https://www.sciencedirect.com/science/article/pii/S014181302100903X   Consider citing these references.

Response 16:

Thank you very much for your question and help us to improve the quality of our manuscript.

Fragrant Woodfern polypeptide sequences similar to Raf (MW524053) were identified using BLAST (E < 1 × 105) (http://blast.ncbi.nlm.nih.gov) [38].

A maximum-likelihood approach was used to construct an unrooted phylogenetic tree based upon amino acid sequences in MEGA 7 (Parameter settings: JTT model, Gamma distribution of site replacement rate, with 1000 Bootstrap reiterations) [39,40].

  1. Tiwari, M.; Yadav, M.; Singh, B.; Pandey, V.; Nawaz, K.; Bhatia, S. Evolutionary and functional analysis of two-component system in chickpea reveals CaRR13, a TypeB RR, as positive regulator of symbiosis. Plant Biotechnology Journal 2021, 19, 2415-2427, doi:10.1111/pbi.13649.
  2. Tiwari, M.; Pandey, V.; Singh, B.; Yadav, M.; Bhatia, S. Evolutionary and expression dynamics of LRR-RLKs and functional establishment of KLAVIER homolog in shoot mediated regulation of AON in chickpea symbiosis. Genomics 2021, 113, 4313-4326, doi:10.1016/j.ygeno.2021.11.022.
  3. Mishra, D.; Suri, G.S.; Kaur, G.; Tiwari, M. Comprehensive analysis of structural, functional, and evolutionary dynamics of Leucine Rich Repeats-RLKs in Thinopyrum elongatum. International Journal of Biological Macromolecules 2021, 183, 513-527, doi:10.1016/j.ijbiomac.2021.04.137.

Point 17:

Line 305: Change 0h to 0 h and check it throughout manuscript.

Response 17:

Thank you very much for your question and help us to improve the quality of our manuscript.

In comparison with the 0 h, the total Rubisco activity, initial Rubisco activity and Rubisco activation state were significantly increased at 0.5 h, but significantly decreased from 1-48 h (Fig. 2a-c); RCA and PRK activities were significantly increased from 0.5-1 h relative to 0 h, but were significantly decreased from 3-48 h (Fig. 2d and Fig. 2e).

Point 18:

Line 309: 3-48 h t. What is t here?

Response 18:

Thank you very much for your question and help us to improve the quality of our manuscript.

In comparison with the 0 h, the total Rubisco activity, initial Rubisco activity and Rubisco activation state were significantly increased at 0.5 h, but significantly decreased from 1-48 h (Fig. 2a-c); RCA and PRK activities were significantly increased from 0.5-1 h relative to 0 h, but were significantly decreased from 3-48 h (Fig. 2d and Fig. 2e).

Point 19:

Line 394: Remove  were

Response 19:

Thank you very much for your question and help us to improve the quality of our manuscript.

To describe the function of DfRaf in tolerance against heat stress, we obtained six independent DfRaf overexpressing Nicotiana tabacum lines (DfRaf-OV-L1-6).

Point 20:

Line 395: Does author mean DfRAf instead of DfRCA

Response 20:

Thank you very much for your question and help us to improve the quality of our manuscript.

To describe the function of DfRaf in tolerance against heat stress, we obtained six independent DfRaf overexpressing Nicotiana tabacum lines (DfRaf-OV-L1-6).

Point 21:

Line 397: semi to Semi

Response 21:

Thank you very much for your question and help us to improve the quality of our manuscript.

Semi-quantitative reverse transcription (RT)-PCR showed that levels of DfRaf transcripts in DfRaf-OV-L2 and DfRaf-OV-L6 was significantly higher.

Point 22:

Line 398: Change to “that levels of DfRaf transcripts in DfRaf-OV-L2 and DfRaf-OV-L6 was significantly higher.

Response 22:

Thank you very much for your question and help us to improve the quality of our manuscript.

Semi-quantitative reverse transcription (RT)-PCR showed that levels of DfRaf transcripts in DfRaf-OV-L2 and DfRaf-OV-L6 was significantly higher.

Point 23:

Line 403, 404: Repetition, remove one of the sentence.

Response 23:

Thank you very much for your question and help us to improve the quality of our manuscript.

Manuscript have been modified

Point 24:

Line 408: remove it in

Response 24:

Thank you very much for your question and help us to improve the quality of our manuscript.

Manuscript have been modified

Point 25:

Line 427: ROS will accumulate rapidly in plant, and high concentrations of ROS will directly affect the normal development of cells. Reference missing consider citing (recommended but not mandatory) https://www.ncbi.nlm.nih.gov/pmc/articles/PMC8784697/

Response 25:

Thank you very much for your question and help us to improve the quality of our manuscript.

Undering abiotic stress, ROS will accumulate rapidly in plant and high concentrations of ROS will directly affect the normal development of cells [49].

  1. Rane, J.; Singh, A.K.; Tiwari, M.; Prasad, P.V.V.; Jagadish, S.V.K. Effective Use of Water in Crop Plants in Dryland Agriculture: Implications of Reactive Oxygen Species and Antioxidative System. Frontiers in Plant Science 2022, 12, doi:10.3389/fpls.2021.778270.

Point 26:

Line 431, 433: Remove it in

Response 26:

Thank you very much for your question and help us to improve the quality of our manuscript.

Manuscript have been modified

Point 27:

Line 559: in-creased to increased

Response 27:

Thank you very much for your question and help us to improve the quality of our manuscript.

Manuscript have been modified

Reviewer 2 Report

Comments to the authors:                                                                                        

MS#genes-1731293-R1

Several points raised by the reviewer were rectified by the authors. Unfortunately, certain key limitations remain in this revised draft, such as:

1. The explanation for DfRaf-OV-L2, DfRaf-OV-L6 plant phenotypic variations in response to high temperature compared to WT and VC lines is not adequately satisfied.If the authors have more replication results/images, they can attach them to this review report. Otherwise, it is not possible to recommend this article for further processing.

2. L 26-27, Please rewrite these lines as the work's final proposal or opinion.

3. Why did the author eliminate the information concerning the important level in L 281-282? This section must be kept, it should not be deleted.

4. L308, 309, 333, 335, 337: putting significant levels in the results section line by line is unnecessary. Because it is already mentioned in the "statistical analysis" section.

5. For Fig. 5 and Fig. 6, the contrast and brightness should be modified.

6. L412, 417,426,431,432,434: In the results section, there is no need to state the significant level (P<0.01) because it is already mentioned in the "statistical analysis" section.

7. L 562-564, please revise these lines as the work's final suggestion or recommendation

Author Response

Response to Reviewer 1 Comments

Point 1:

Several points raised by the reviewer were rectified by the authors. Unfortunately, certain key limitations remain in this revised draft, such as:

  1. The explanation for DfRaf-OV-L2, DfRaf-OV-L6 plant phenotypic variations in response to high temperature compared to WT and VC lines is not adequately satisfied. If the authors have more replication results/images, they can attach them to this review report. Otherwise, it is not possible to recommend this article for further processing.

Response 1:

Thank you very much for your question and help us to improve the quality of our manuscript.

To describe the function of DfRaf in tolerance against heat stress, we obtained six independent DfRaf overexpressing Nicotiana tabacum lines (DfRaf-OV-L1-6). DfRaf gene in these lines were detected by genomic DNA PCR, the results showed that DfRaf-OV-L1-6 were PCR positive. Semi-quantitative reverse transcription (RT)-PCR showed that levels of DfRaf transcripts in DfRaf-OV-L2 and DfRaf-OV-L6 was significantly higher. So DfRaf-OV-L2 and DfRaf-OV-L6 were chosen for phenotype analysis. Under normal conditions, there was no significant difference in phenotypes between the DfRaf-OV-L2, DfRaf-OV-L6, VC and WT plants; compared with the leaves of DfRaf-OV-L2 and DfRaf-OV-L6, the leaves of WT and VC have turned yellow under high temperature treatment (Fig. 7).

Fig. 7 Genomic DNA PCR and semi-quantitative reverse transcription (RT)-PCR identification of transformation with DfRaf and phenotype of tobacco transformed with the DfRaf gene on the heat treatment. (a) PCR identification of transformation with DfRaf. M: Trans2K DNA Marker, -: ddH2O negative control, +: Positive control, VC: vector control, WT: wild type; and DfRaf overexpression lines DfRaf-OV-L1-6; (b) semi-quantitative reverse transcription (RT)-PCR identification of transformation with DfRaf. WT stands for wide-type; DfRaf-OV-L1-6 are transformation with the DfRaf gene; (c) Phenotype of tobacco transformed with the DfRaf gene on the heat treatment. WT stands for Wide-type; DfRaf-OV-L2 and DfRaf-OV-L6 were transformed with the DfRaf gene; (d) Phenotype of leaves of WT, VC, DfRaf-OV-L2 and DfRaf-OV-L6 under high temperature treatment.

Point 2:

  1. L 26-27, Please rewrite these lines as the work's final proposal or opinion.

Response 2:

Thank you very much for your question and help us to improve the quality of our manuscript.

The results showed that overexpression of DfRaf gene increased the Rubisco activity, which enhance high temperature tolerance of plants.

Point 3:

  1. Why did the author eliminate the information concerning the important level in L 281-282? This section must be kept, it should not be deleted.

Response 3:

Thank you very much for your question and help us to improve the quality of our manuscript.

If I find this L 281-282 wrong, can you copy it for me.

Relative to 0 h, the relative expression of DfRCA in Fragrant Woodfern leaves was significantly higher than 0 h at 0.5 – 3 h but significantly lower than 0 h at 6 - 48 h (Fig. 3d); the relative expression of DfPRK in leaves of Fragrant Woodfern was significantly higher than 0 h at 0.5 h but significantly lower than 0 h at 3 – 48 h (Fig. 3e). From the above qRT-PCR analysis, high temperature treatment upregulated the expression of DfRaf, which was significantly higher than that of the other genes. So, DfRaf was selected for the further study.

Point 4:

  1. L308, 309, 333, 335, 337: putting significant levels in the results section line by line is unnecessary. Because it is already mentioned in the "statistical analysis" section.

Response 4:

Thank you very much for your question and help us to improve the quality of our manuscript.

Manuscript have been modified.

Point 5:

  1. For Fig. 5 and Fig. 6, the contrast and brightness should be modified.

Response 5:

Thank you very much for your question and help us to improve the quality of our manuscript.

Fig. 5 and Fig. 6 have been modified.

Fig. 5 Phylogenetic tree and conserved Motif analyses for the DfRaf protein. A maximum-likelihood approach was used to construct a phylogenetic tree based upon amino acid sequences using MEGA 7 (Parameter settings: JTT model, Gamma distributions of site replacement rate, with 1,000 Bootstrap replicate interations).

Fig. 6 Multiple sequence alignment for the DfRaf protein.

Point 6:

  1. L412, 417,426,431,432,434: In the results section, there is no need to state the significant level (P<0.01) because it is already mentioned in the "statistical analysis" section.

Response 6:

Thank you very much for your question and help us to improve the quality of our manuscript.

Manuscript have been modified.

Point 7:

  1. L 562-564, please revise these lines as the work's final suggestion or recommendation

Response 7:

Thank you very much for your question and help us to improve the quality of our manuscript.

The results showed that overexpression of DfRaf gene increased the Rubisco activity, which enhance high temperature tolerance of plants.

Round 3

Reviewer 2 Report

Comments to the author,

The author revised all points made by the reviewer. So, it can be considered for further processing.